# WOMD-Reasoning: A Large-Scale Dataset for Interaction Reasoning in Driving

Yiheng Li [* 1]   Cunxin Fan [* 1]   Chongjian Ge [1]   Seth Z. Zhao [2]   Chenran Li [1]   Chenfeng Xu [1]   Huaxiu Yao [3]
Masayoshi Tomizuka [1]   Bolei Zhou [2]   Chen Tang [1 4 †]   Mingyu Ding [1 3 †]   Wei Zhan [1]

## Abstract

Language models uncover unprecedented abilities in analyzing driving scenarios, owing to their limitless knowledge accumulated from text-based pre-training. Naturally, they should particularly excel in analyzing rule-based interactions, such as those triggered by traffic laws, which are well documented in texts. However, such interaction analysis remains underexplored due to the lack of dedicated language datasets that address it. Therefore, we propose **W**aymo **O**pen **M**otion **D**ataset-**Reasoning** (**WOMD-Reasoning**), a comprehensive large-scale Q&As dataset built on WOMD focusing on describing and reasoning traffic rule-induced interactions in driving scenarios. WOMD-Reasoning also presents by far the largest multi-modal Q&A dataset, with 3 million Q&As on real-world driving scenarios, covering a wide range of driving topics from map descriptions and motion status descriptions to narratives and analyses of agents' interactions, behaviors, and intentions. To showcase the applications of WOMD-Reasoning, we design Motion-LLaVA, a motion-language model fine-tuned on WOMD-Reasoning. Quantitative and qualitative evaluations are performed on WOMD-Reasoning dataset as well as the outputs of Motion-LLaVA, supporting the data quality and wide applications of WOMD-Reasoning, in interaction predictions, traffic rule compliance plannings, etc. The dataset and its vision modal extension are available on https://waymo.com/open/download/. The codes & prompts to build it are available on https://github.com/yhli123/WOMD-Reasoning.

---
[*]Equal contribution  [1]UC Berkeley [2]UCLA [3]UNC-Chapel Hill [4]UT Austin. Correspondence to: Mingyu Ding <md@cs.unc.edu>, Chen Tang <chen.tang@austin.utexas.edu>.

*Proceedings of the $42^{nd}$ International Conference on Machine Learning*, Vancouver, Canada. PMLR 267, 2025. Copyright 2025 by the author(s).

## 1. Introduction

Language-guided models improve explainability, controllability, and performance of autonomous driving tasks (Touvron et al., 2023; Brown et al., 2020; Chen et al., 2023; Fu et al., 2023; Malla et al., 2023; Xu et al., 2024a; Li et al., 2024a; Arai et al., 2024; Park et al., 2024; Li et al., 2024b; Cui et al., 2024; Ma et al., 2024; Yuan et al., 2024; Greer & Trivedi, 2024; Ding et al., 2024a; Han et al., 2024; Wang et al., 2024b), owing to their enormous text-based knowledge acquired through the pre-training. To help language models elevate abilities on driving-related tasks, various language-based datasets (Sima et al., 2023; Qian et al., 2024; Kim et al., 2018; Sachdeva et al., 2024a; Malla et al., 2023; Inoue et al., 2024; Zhou et al., 2024; Tian et al., 2024b; Arai et al., 2024; Park et al., 2024; Li et al., 2024b; Ma et al., 2024) have been proposed to fine-tune these LLMs. However, due to the complexity and variations in summarizing traffic rule-based interactions in real driving scenarios, existing datasets often focus on interactions based on spatial proximity. For example, in BDD-X (Kim et al., 2018), interactions are often limited to the spatial blocking, like *'The car moves back into the left lane because the school bus in front of it is stopping.'*; while in another example, DriveLM (Sima et al., 2023) attributes interaction to *'keep a safe distance'* without further detailed explanations. Likewise, most existing works only cover interactions induced by proximity, as shown in our analysis in Table 1. While reasoning about many non-proximate interactions, like those caused by traffic rules, are vital for the safety-critical decision-making in autonomous driving systems (Wang et al., 2022; Zhang et al., 2022; Zhao et al., 2024; Ding & Zhao, 2023; Roelofs et al., 2022), these interactions are rarely included in previous datasets, leading to suboptimal performance of LLMs fine-tuned on these datasets. Additionally, most datasets in driving are not large enough to support multi-modal fine-tuning across diverse tasks such as scene description, prediction, and planning.

For the sake of interaction descriptions and reasoning, this work introduces WOMD-Reasoning dataset, a large-scale multi-modal dataset centered on language Q&As based on WOMD (Ettinger et al., 2021). To incorporate interactions induced by traffic rules and human intentions, we build an automated data curation pipeline by prompting ChatGPT

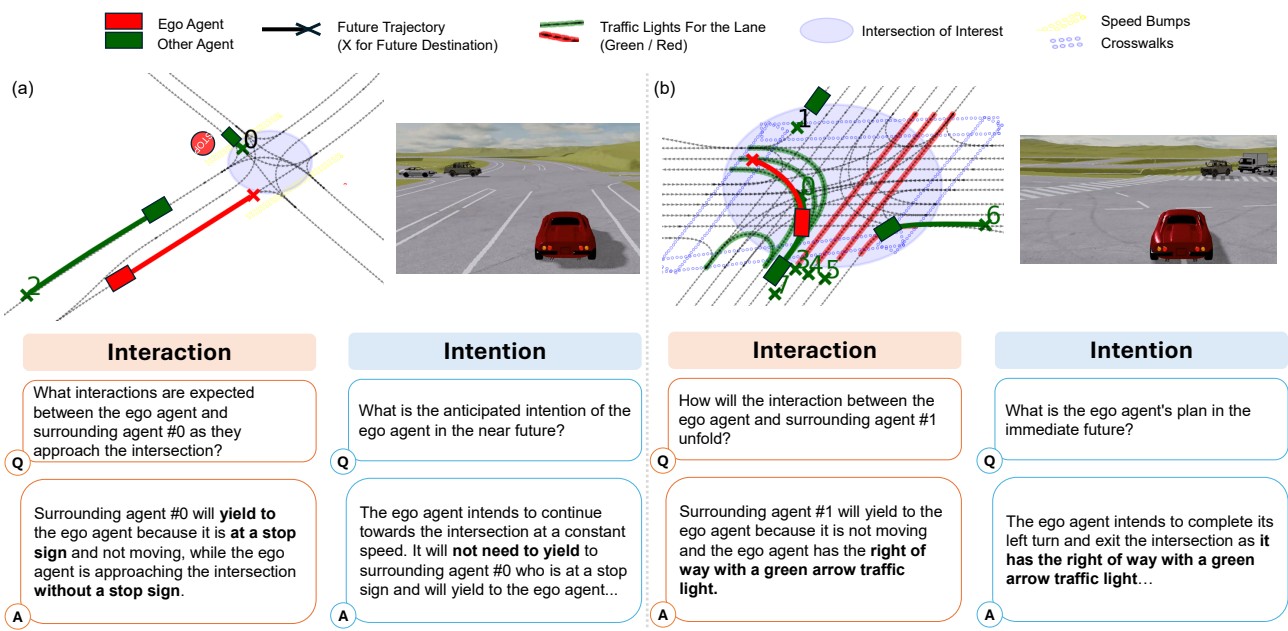

*Figure 1.* **Examples of traffic rule-induced interactions in WOMD-Reasoning dataset.** (a) captures the traffic rule-induced interaction between the ego agent and agent #0, attributing it correctly to the stop signs. (b) shows the traffic light-controlled yielding interaction between the ego agent and agent #1. The front-view visualization is created by MetaDrive simulator (Li et al., 2022)

(Brown et al., 2020; OpenAI et al., 2024) with a set of well-designed prompts. Together with a rule-based translator to convert initial motion data into language, we create an automated pipeline to generate the language dataset and achieve an approximate accuracy rate of 90%. WOMD-Reasoning dataset covers interactions induced by traffic rules and human intentions with 409k Q&As, like those shown in Figure 1. As indicated in Table 1, WOMD-Reasoning contains the highest number of total Q&As and interaction-specific Q&As, supporting a wide range of applications, including scene description, prediction and planning. Figure 2 further demonstrates its uniqueness of interaction descriptions in WOMD-Reasoning with vocabulary statistics. In addition to comprehensive and in-depth interactions, WOMD-Reasoning is also the largest language dataset for real-world driving data, to the best of our knowledge and by its publication. Additionally, to facilitate multi-modal driving-related tasks like training end-to-end driving models, in addition to the LiDAR (Chen et al., 2024) info provided by Waymo, we render both the corresponding bird's eye view and front view videos of covered driving scenarios by the MetaDrive simulator (Li et al., 2022), which are attached to the dataset.

To showcase the applications of WOMD-Reasoning, we fine-tune our Motion-LLaVA, which is built upon LLaVA (Liu et al., 2024), on WOMD-Reasoning. With careful design and a set of effective training and inference strategies ablated, Motion-LLaVA achieves high accuracy in both driving scenario understanding and interaction analysis, which unequivocally advocate the effectiveness of WOMD-Reasoning in boosting the interaction prediction abilities of

driving language models.

Our contributions can be summarized as follows:

- We propose WOMD-Reasoning, the largest multi-modal dataset with 3 million Q&A pairs centered on interaction reasoning in driving. It provides extensive insights into critical but previously overlooked interactions induced by traffic rules and human intentions. Quantitative and qualitative evaluations verify its wide coverage in interaction descriptions and reasoning, as well as its accuracy.

- Fine-tuned with WOMD-Reasoning, our Motion-LLaVA is capable of providing detailed and insightful interaction predictions on various driving scenarios, supporting the effectiveness of WOMD-Reasoning. Extensive evaluations with various metrics validate the wide coverage and high quality of the answers of Motion-LLaVA fine-tuned on WOMD-Reasoning, ranging from driving scenario descriptions to traffic rule-compliant planning.

- To optimize the performance of Motion-LLaVA, a motion-language model designed for WOMD-Reasoning with robust capabilities, we benchmark task performance across various configurations, including input modalities, reasoning techniques, and network architectures.

*Table 1.* **Comparison between WOMD-Reasoning Dataset and Previous Real-world Language Datasets for Driving.** WOMD-Reasoning covers more comprehensive kinds of interactions, with a significantly higher size supporting a wide range of applications.

| Dataset | Data Source | Statistics | | | Interactions | | | Applications | | |
|---|---|---|---|---|---|---|---|---|---|---|
| | | Total Scenes | Total Q&As | Interaction Q&As | Distance-induced | Traffic Rule-induced | Human Intention-induced | Scene Descriptions | Motion Prediction | Motion Planning |
| nuScenes-QA(Qian et al., 2024) | nuScenes(Caesar et al., 2020) | 34k | 460k | 0 | ✗ | ✗ | ✗ | ✓ | ✗ | ✗ |
| BDD-X(Kim et al., 2018) | BDD(Xu et al., 2017) | 26k | 26k | 26k | ✓ | ✗ | ✗ | ✓ | ✗ | ✓ |
| DriveLM(Sima et al., 2023) | nuScenes(Caesar et al., 2020) | 5k | 443k | ≈199k | ✓ | ✗ | ✗ | ✓ | ✓ | ✓ |
| Rank2Tell(Sachdeva et al., 2024a) | Rank2Tell(Sachdeva et al., 2024a) | 116 | >116 | >116 | ✓ | ✓ | ✓ | ✓ | ✓ | ✓ |
| DRAMA(Malla et al., 2023) | DRAMA(Malla et al., 2023) | 18k | 102k | <18k | ✓ | ✓ | ✗ | ✓ | ✗ | ✓ |
| NuInstruct(Ding et al., 2024b) | nuScenes(Caesar et al., 2020) | 850 | 91k | <46k | ✓ | ✗ | ✗ | ✓ | ✓ | ✓ |
| nuCaption(Yang et al., 2025) | nuScenes(Caesar et al., 2020) | <1k | 420k | ≈140k | ✓ | ✗ | ✗ | ✓ | ✗ | ✓ |
| Tod3Cap(Jin et al., 2024) | nuScenes(Caesar et al., 2020) | 850 | ≈2,300k | 0 | ✗ | ✗ | ✗ | ✓ | ✗ | ✗ |
| **WOMD-Reasoning** | WOMD (Ettinger et al., 2021) | **63k** | **2,940k** | **409k** | ✓ | ✓ | ✓ | ✓ | ✓ | ✓ |

## 2. Related Work

**Application of Language Modal in Autonomous Driving.** In recent years, language has been widely explored in autonomous driving, as incorporating language into implicit features could enhance the explainability (Yang et al., 2023; Roh et al., 2020; Sriram et al., 2019; Chen et al., 2019; Mao et al., 2023a; Zhang et al., 2024). For example, Kuo et al. (Kuo et al., 2022) employ language features in the trajectory prediction model, ensuring both explainability and consistency between language and predictions. Similarly, Zhong et al. (Zhong et al., 2023) use language input and LLMs to generate conditions in diffusion models, thereby controlling the generation of driving scenarios. Recently, general driving agents have been developed (Xu et al., 2024a; Chen et al., 2023; Sima et al., 2023; Mao et al., 2023b), aiming to integrate all functions—perception, prediction, and planning—into a single language agent. However, we observe that their performance in interaction analysis is not entirely satisfying, especially in non-proximal interactions like those induced by traffic rules. This shortcoming is largely attributed to the lack of interaction analysis in language datasets specific to driving scenarios.

**Language Datasets in Autonomous Driving.** Language datasets for driving scenarios have recently been developed to support related LLM-based work. Due to the controllability of simulations, several simulation-based datasets have been created (Chen et al., 2023; Sima et al., 2023). However, simulations often fail to capture real-world interactions. To address this, recent studies (Kim et al., 2018; Sachdeva et al., 2024a; Malla et al., 2023; Wang et al., 2024a) incorporate human labeling of real-world driving scenarios to ensure the inclusion of genuine interactions. Nevertheless, obtaining high-quality human labels is labor-intensive and costly, limiting the size and coverage of these datasets. Moreover, interactions driven by traffic rules and human intentions have often been overlooked. To streamline the labeling process, some studies (Deruyttere et al., 2019; Nie et al., 2023; Qian et al., 2024; Sima et al., 2023) have generated rule-based labels, although these typically cover only basic language elements like scene descriptions, leaving interaction analysis to human labelers. To minimize human labor, we first employ manual rules and then utilize ChatGPT-4 (OpenAI et al., 2024) to build our dataset automatically. The resulting dataset, WOMD-Reasoning, is rich in interaction details, particularly those stemming from traffic rules and human intentions. The designed automated data-curation pipeline enables the dataset to be significantly larger. We also create a high-quality human-verified subset. Preliminary human evaluation provides positive feedback with an approximate accuracy of 90%.

**Fine-tuning LLM for Driving Tasks.** Recent years have witnessed a surge of research on leveraging LLMs and vision-language models (VLMs) for driving applications (Seff et al., 2023; Chen et al., 2023; Jin et al., 2023; Xu et al., 2024b; Sima et al., 2023; Shao et al., 2023; Mao et al., 2023a; Tian et al., 2024a). However, handling heterogeneous inputs such as vectorized motion representation remains challenging due to their domain gaps to most existing VLMs and LLMs. Notable prior work, such as (Mao et al., 2023a), designed natural language prompt structures to accommodate numerical motion data, while another approach (Chen et al., 2023) proposed training LLMs with custom-built motion vector encoders.

Our Motion-LLaVA approach is designed for benchmarking the performance of WOMD-Reasoning. Inspired by the design of (Chen et al., 2023), Motion-LLaVA augmenting existing LLM architectures with tailored components. We employ a motion prediction model as a motion vector encoder, utilizing its prior knowledge to streamline the training pipeline. To enhance reasoning accuracy, we adopt a Chain-of-Thought (CoT) (Wei et al., 2022) approach, allowing the model to base its reasoning on factual grounding, mitigating potential hallucinations. Based on WOMD-Reasoning, our Motion-LLaVA demonstrates the practicality of applying VLMs to real-world vectorized motion data to accurately answer diverse driving-related questions correctly.

## 3. Method

In this part, we first introduce how to build the WOMD-Reasoning dataset. To verify the effectiveness of WOMD-

Reasoning, we introduce Motion-LLaVA, a multi-modal fine-tuning approach, to build motion language models upon WOMD-Reasoning. The Motion-LLaVA method as well as the fine-tuning process are also included here.

### 3.1. Building WOMD-Reasoning Dataset

Manually constructing autonomous driving datasets with fine-grained natural-language labels regarding the interactions among road participants would involve intense human labor, which is one of the main reasons why previous datasets have insufficient interaction analysis and a very limited overall size (Sachdeva et al., 2024b). To reduce human labor while obtaining reasonably useful data on vehicle interactions, we propose a fully automatic data-curation pipeline to label the WOMD dataset with language: We first develop a rule-based program to interpret the motion dataset, which contains trajectories and the HD map, into language; and then build a set of prompts to utilize the reasoning abilities of ChatGPT-4 (OpenAI et al., 2024) to generate interaction analysis and reasoning, and organize the results into the target Q&A format. Details of each step are attached in the Appendix A.1. Codes for interpreting the motion dataset as well as the full prompts for ChatGPT-4 are available in supplemental materials.

Our analysis is performed on Microsoft Azure ChatGPT-4-Turbo API, which costs around 12,750 USD. The details of WOMD-Reasoning will be provided in Part 4.

### 3.2. Fine-tuning Multi-modal Model on WOMD-Reasoning

To showcase the application of WOMD-Reasoning dataset, we use our Motion-LLaVA approach to fine-tune upon LLaVA (Liu et al., 2024). LLaVA is chosen to accommodate motion data and text prompts input at the same time, with the vision encoder adapted to a motion data encoder. Inspired by (Shao et al., 2023), we utilize encoders from motion prediction models MultiPath++ (Varadarajan et al., 2021) for encoding motion data. Ablations in Table 15 prove the usefulness of this encoder. Some detailed fine-tuning strategies are in Appendix A.10. Details on how we process agent IDs in Motion-LLaVA are listed in Appendix A.3. The overall Motion-LLaVA pipeline is illustrated in Figure 4.

During training, we unfreeze all components including the motion vector encoder, and train on all Q&A pairs in WOMD-Reasoning simultaneously to avoid the potential catastrophic forgetting. To prevent Motion-LLaVA from forgetting its inherent LLM's knowledge, we mix up a few questions irrelevant to the current driving scenario into the training process (Zhai et al., 2023). During inference, we adopt Chain-of-Thought (CoT) Reasoning strategy (Wei et al., 2022) to reduce the hallucination. Specifically, we first prompt Motion-LLaVA to answer the **Factual** questions, which include questions in map environment, ego, and other agents' motion status. Then, we aggregate these answers and feed them into Motion-LLaVA as the contexts for answering **Reasoning** questions in interactions and intentions analysis. Examples of these contexts can be found in the Appendix A.8. We ablate in Part A.10 that such a CoT strategy can effectively improve the quality of answering reasoning questions.

Specifically, we take LLaVA-v1.5-7b (Liu et al., 2024) as the pre-trained VLM. Since MultiPath++ (Varadarajan et al., 2021) did not release their codes nor checkpoints, we take the implementations by (Konev, 2022). Our multi-modal fine-tuning takes 2 GPU days ($\approx$ 1 day on 2xNVIDIA A6000 GPUs) to train 1 epoch on the entire training set of WOMD-Reasoning. For quantitative evaluations, we randomly select $\approx$ 1,000 QA pairs from the validation set of WOMD-Reasoning.

To benchmark Motion-LLaVA model, we utilize a suite of standard language metrics, including BLEU-4 (Papineni et al., 2002), METEOR (Banerjee & Lavie, 2005), CIDEr (Vedantam et al., 2015), SPICE (Anderson et al., 2016), and ROUGE-L (Lin, 2004), along with GPT Score (Radford et al., 2018) to assess the quality of the generated content. Details on the prompts used to calculate the GPT Score can be found in Appendix A.5.

## 4. WOMD-Reasoning Dataset Specifications

In this section, we provide detailed information about WOMD-Reasoning. We first present the organization and statistics of the dataset, then provide quality analysis based on case studies and preliminary human evaluations. Finally, we compare our dataset to previous ones to show the strengths of WOMD-Reasoning in diverse interactions, huge size, and wide potential applications.

### 4.1. Dataset Statistics

Table 2 provides the statistics of Q&As in WOMD-Reasoning dataset, as well as in every Q&A class. Our Q&As can be classified into two parts: The **Factual** contents and the **Reasoning** contents, where the first part serves as the context for the second part. The factual contents contain three topics: *Map Environment*, which includes the existence and category of intersections, existence and places of stop signs and crosswalks, and count of lanes, etc. *Ego Agent's Motion Status* and *Other (Surrounding) Agents' Motion status*, which describe each involved agent's speed, acceleration, direction, related traffic light and sign, and relative position to the intersection center as well as to the ego agent. The factual descriptions alone are also good for training language scene describers.

*Table 2.* Sizes of WOMD-Reasoning dataset.

| Set | Map | Ego Agent | Other Agents | Interactions | Intentions | Total | Scenes |
|---|---|---|---|---|---|---|---|
| Training | 188k | 268k | 1,635k | 287k | 52k | 2,430k | 52k |
| Validation | 41k | 58k | 341k | 59k | 11k | 510k | 11k |
| Total | 229k | 326k | 1,976k | 346k | 63k | 2,940k | 63k |
| Per Scene Ave. | 3.58 | 5.09 | 30.88 | 5.41 | 0.98 | 45.95 | - |

*Table 3.* Human Evaluations of WOMD-Reasoning Dataset.

| Q&A Class | All | Interactions | Intentions |
|---|---|---|---|
| Accuracy | 91.99% | 91.03% | 87.50% |

Our key part, the Reasoning contents, first comes with Q&As on *Interactions*, which includes a summary of interactions between each surrounding agent and the ego agent and the reasoning behind such interactions. Then by summing up all interaction information, we offer the *Intentions*, which give the predicted intention of the ego agent, considering its responses to all the interactions. The interaction and intention reasoning parts provide an unprecedented tool for training language-based prediction and planning agents.

As shown in Table 1, WOMD-Reasoning contains **346k** unique traffic rule-induced and intention-induced interaction Q&As, as well as **63k** comprehensive intention predictions of the ego agent containing responses to the interactions, which rarely exist in previous datasets. We will show these features through examples and statistics in the following section. Furthermore, we also claim that to the best of our knowledge, WOMD-Reasoning is the largest real world language data set for driving, since our WOMD-Reasoning provides roughly **3 million** Q&A pairs for **63k** scenes in WOMD, which are 6 - 113 times greater than those of other datasets we compare to.

Compared with existing data sets, as shown in Table 1, our WOMD-Reasoning is the **largest** real-world dataset in terms of total scenes, total Q&As, and the total interactions included. Also, WOMD-Reasoning supports Q&As in scene descriptions, prediction, and planning. Together with the simulated video of each covered scenario, it is suitable for training VLMs to perform nearly all major tasks in autonomous driving. Diving even deeper into the interaction reasoning in these datasets, we summarize that most interactions covered in previous datasets are caused by a very near distance. However, as we show in Sec. 4.3, many interactions happen when the two counterparts are initially far away, but they establish interactions by following the traffic rules or by human intentions. Our WOMD-Reasoning contains a wealth of this interaction information, as shown in Figure 2, the vocabulary statistics of the interaction and intention part of our dataset.

### 4.2. Interaction Contents

To testify to the quality of WOMD-Reasoning, we show examples from it containing rich traffic rule-induced and

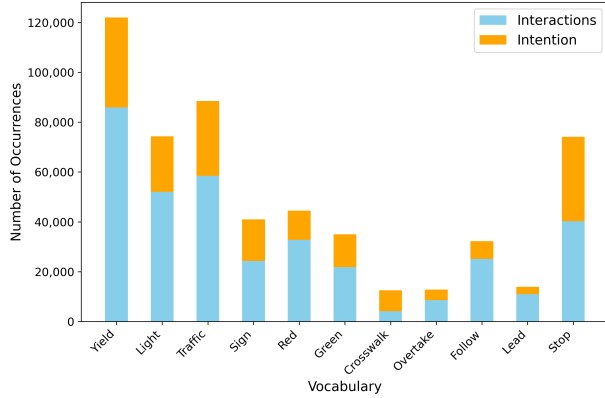

*Figure 2.* **Selected vocabulary statistics in WOMD-Reasoning.** We stat vocabularies strongly related to traffic rule-induced and human intention-induced interactions in WOMD-Reasoning, illustrating that it contains abundant such interaction descriptions and reasoning.

human intention-induced interactions.

**WOMD-Reasoning contains traffic rule-induced interaction information.** Firstly, in Figure 1 (a), we show a traffic scenario of an intersection controlled by the stop sign. In this case, the agent #0 is quite far away from the ego agent at the starting (current) moment, thus their interactions would rarely be considered in previous datasets, as they mainly cover distance-induced interactions. However, the two agents do have a significant interaction: Controlled by the stop signs, the agent #0 has to yield to the ego agent, even though the ego agent is still far from the intersection. This case vividly proves the remarkable importance of the traffic rule-induced interactions we cover.

Another example is a traffic light-controlled intersection shown in Figure 1 (b). Similar to the previous one, agent #0 is initially not close to the ego agent. However, thanks to our analysis of traffic rules, our language dataset effectively captures the interaction that the agent #1 would yield to the ego agent who is turning left and having right-of-way due to an arrow green light. This further proves that traffic rules are essential in determining the interactions.

**WOMD-Reasoning contains human intention-induced interaction information.** Beside traffic rules, our dataset also covers human intention-induced interactions. We provide an additional example to show this, as well as to demonstrate a whole picture of each section of the dataset. In Figure 3, WOMD-Reasoning provides fruitful information on the human intention-induced interaction - overtaking, based on pattern recognition abilities built in the GPT. We also show 1-2 Q&As for each category of the dataset for completeness.

### 4.3. Dataset Quality Evaluation by Human

To grab a sense of the accuracy of WOMD-Reasoning, we execute a human evaluation process. Consent was obtained

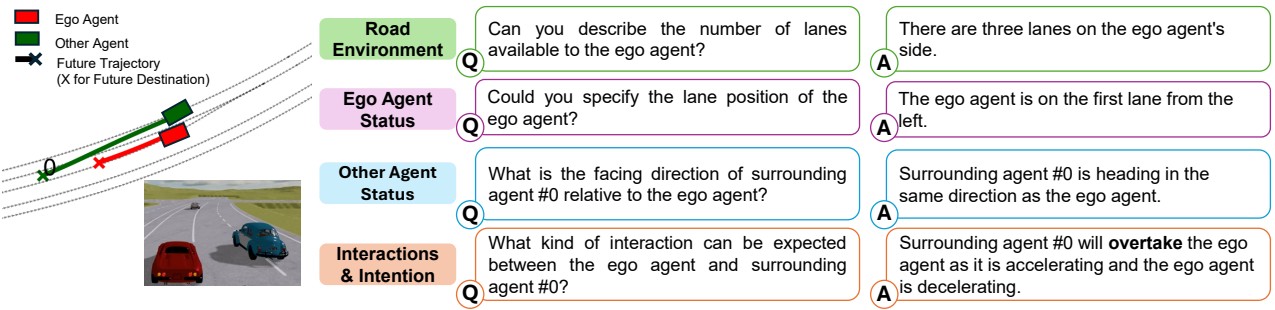

*Figure 3.* **A demonstration of Q&As in each part of WOMD-Reasoning dataset.** We show Q&As in all categories regarding the scenario while demonstrating language analysis of overtaking, a human intention-induced interaction in WOMD-Reasoning dataset.

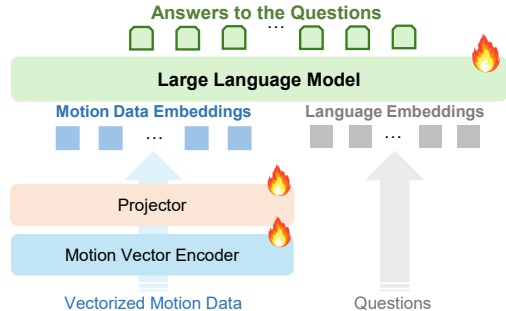

*Figure 4.* **The Motion-LLaVA pipeline fine-tuning a multi-modal model with WOMD-Reasoning.** Motion data go through pre-trained motion vector encoders from Multipath++ (Varadarajan et al., 2021) and a projector layer to serve together with the questions in WOMD-Reasoning as the inputs. The answers in WOMD-Reasoning serve as the supervision.

from all the people interviewed. We ask interviewers to evaluate whether the correct answer to the question is included in the answer provided in the dataset. We invite 4 people to judge 1,610 Q&As in total. The results are shown in Table 3. We conclude that WOMD-Reasoning achieves a satisfying accuracy, especially considering it is automatically generated. Further human-based data cleaning is ongoing.

### 4.4. Vision Extension with Simulations

To support applications requiring visual information, we provide an API to extend WOMD-Reasoning scenario descriptions to the vision domain, compensating for the absence of raw camera data in the original WOMD dataset. We achieve this by utilizing an open-source traffic scenario simulation platform, namely ScenarioNet (Li et al., 2023), and employing the MetaDrive (Li et al., 2022) simulator to replay WOMD-Reasoning scenarios in both BEV layout and ego-view images through 3D rendering. This approach enables visual inputs to support the application of WOMD-Reasoning for future research in the vision and language domain. For video generation, we produce a 90-frame 10 Hz video aligned with Waymo's trajectory data, including 10 frames of historical footage and 80 frames of future trajectory. Users may choose to use the historical portion for

prediction tasks.

## 5. Dataset Application and Evaluation with Motion-LLaVA

To demonstrate a real-world application of WOMD-Reasoning, we design Motion-LLaVA, a multi-modal model taking motion data as input, and fine-tune it on WOMD-Reasoning. We show that the model gains significant abilities from WOMD-Reasoning in predicting interactions, especially those caused by traffic rules, and in planning right-of-way compliant route for the ego. Furthermore, its abilities extend to answer various questions related to driving across factual and reasoning categories, which is supported by both qualitative showcases and quantitative evaluations on language metrics. These highlight the broad applications of WOMD-Reasoning for the autonomous driving industry.

### 5.1. Interaction Prediction

Since WOMD-Reasoning is proposed to focus on the interaction analysis, one of the key abilities of Motion-LLaVA fine-tuned on it is to predict interactions, especially the traffic rule-induced interactions. We present a few interaction prediction examples answered by Motion-LLaVA in Figure 5. These results confirm that by fine-tuning on WOMD-Reasoning, Motion-LLaVA unleashes the potential of language models for predicting traffic rule-related interactions in unseen real-world scenarios.

### 5.2. Traffic Rule Compliant Planning

Based on thorough interaction analysis, traffic rule compliant planning can naturally be achieved by summarizing interactions, as we do in producing ego agents' intentions in WOMD-Reasoning. Motion-LLaVA also perfectly learns this ability. In Figure 6, we demonstrate that Motion-LLaVA fine-tuned on WOMD-Reasoning can offer traffic rule-compliant future trajectory plans, which clearly extends the applications of WOMD-Reasoning into planning.

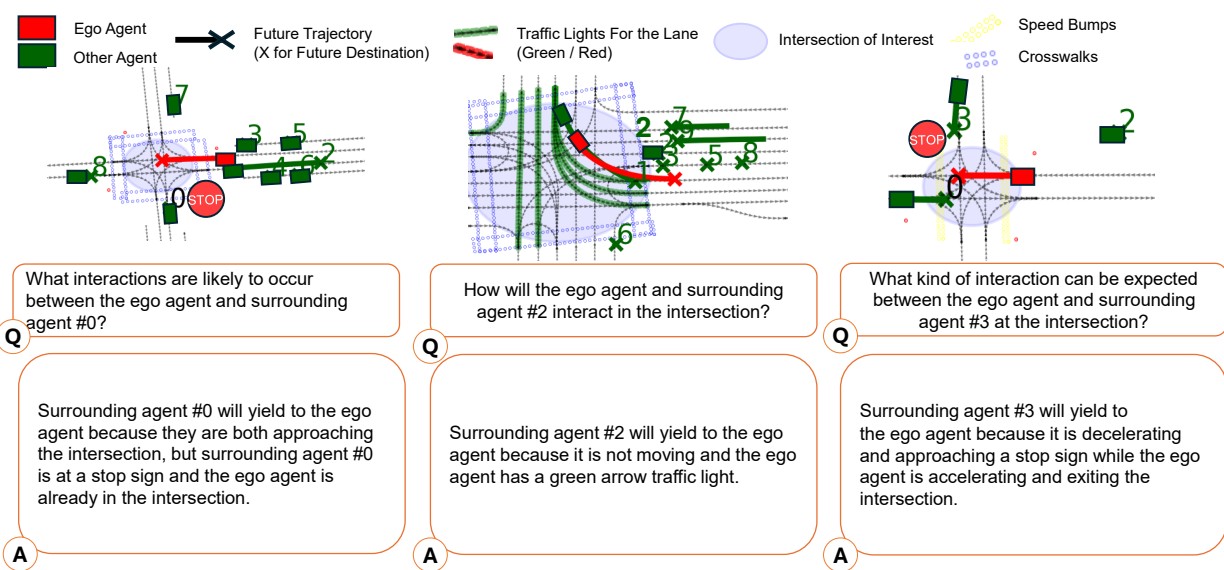

*Figure 5.* **Interaction predictions made by Motion-LLaVA on various WOMD scenarios.**

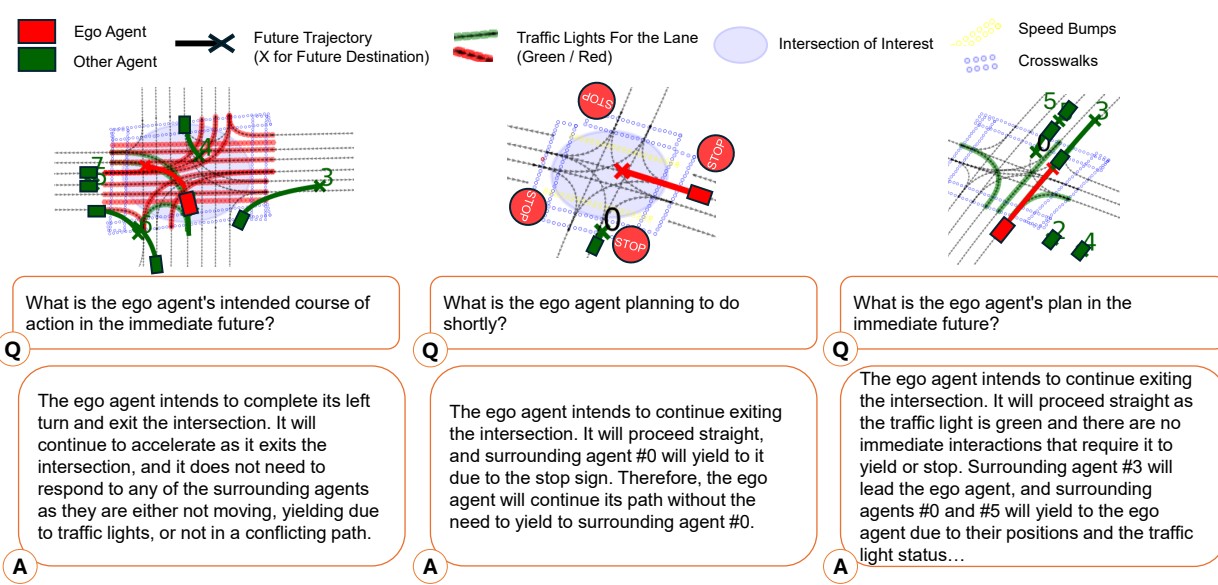

*Figure 6.* **Traffic rule-compliant planning made by Motion-LLaVA on various WOMD scenarios.**

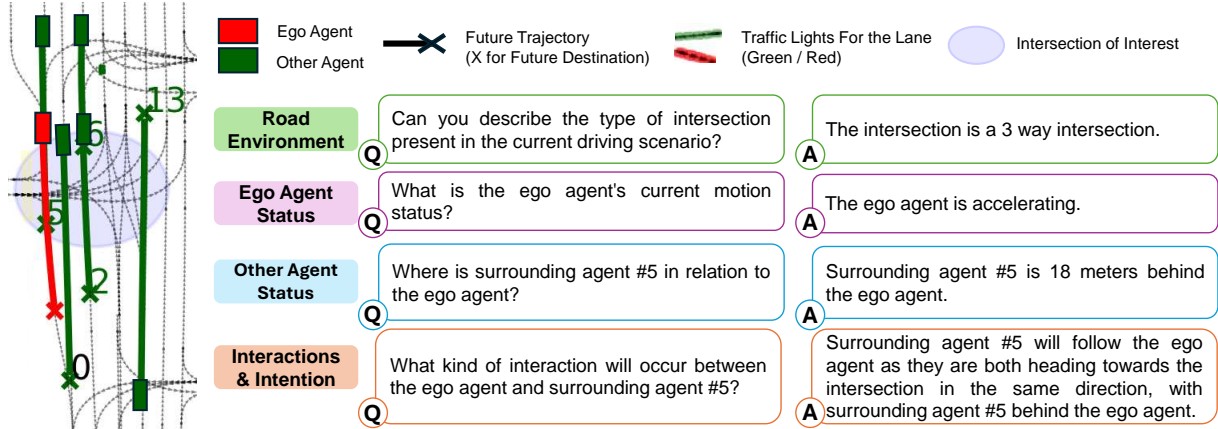

Figure 7. **An example set of answers to diverse driving-related questions generated by Motion-LLaVA on a WOMD scenario.** We see that Motion-LLaVA is capable of answering all topics of questions covered by WOMD-Reasoning accurately.

### 5.3. Answering Various Driving-related Questions

We further present a set of Q&A examples answered by Motion-LLaVA covering each category of WOMD-Reasoning in one specific driving scenario, shown in Figure 7. We observe that Motion-LLaVA model is capable of providing accurate narratives and analysis on diverse driving-related contents thanks to the wide coverage of WOMD-Reasoning.

### 5.4. Validation of Motion-LLaVA

To validate the effectiveness of Motion-LLaVA and the fine-tuning process, we present evaluation metrics of all generated answers, as well as those in the two categories, i.e. the **Facts** which include the road environment and agents' motion status, and the **Reasonings** which cover analysis on interactions and intentions. Benchmark results are shown in Table 4. We first verify the effectiveness of these language metrics with shuffled ground-truths, which attach shuffled true answers to questions randomly. Results confirm that low scores are granted for this model, indicating that a similar structure with wrong answer content would not be able to cheat our metrics. Then we benchmark the off-the-shelf LLaVA (Liu et al., 2024), where BEV plots are used as the vision input for the driving scenario. This non-fine-tuned version earns a very low score in all metrics, confirming that LLaVA itself cannot answer driving-related questions correctly. Then, we perform fine-tuning on WOMD-Reasoning. We compare two models with different modalities as motion input: LLaVA using the BEV plot of the driving scene as the vision input (Liu et al., 2024), and Motion-LLaVA which uses motion data as input. We observe that both fine-tuned multi-modal models significantly outperform previous baselines across all metrics and question types, underscoring the value of fine-tuning on WOMD-Reasoning for addressing driving-related queries. Compared to the fine-tuned LLaVA, Motion-LLaVA achieves superior performance across all

metrics and question types, highlighting the advantages of utilizing the motion data as input. We hypothesize that this performance gap is due to the information loss that occurs when converting motion data into a BEV plot with a scale bar. We also visualize and compare the answers of Motion-LLaVA and these baselines on the same driving scene in Appendix A.6.

In addition to language metrics, we also provide objective metrics to the answers to questions regarding **Facts**. We report the root median squared error (Median Error), and accuracy within a 1.0-meter tolerance ($Acc_{1.0}$) in Table 5, following similar metrics used in (Jin et al., 2023). Results confirm that Motion-LLaVA is capable of providing numerically accurate answers to factual questions.

Table 5. Accuracy of Motion-LLaVA answers to factual questions.

| Model | $Acc_{1.0}$ (↑) | Median Error (↓) |
|---|---|---|
| Fine-tuned LLaVA (Liu et al., 2024) | 10.9% | 5.0 |
| Motion-LLaVA (Ours) | **56.2%** | **1.0** |

## 6. Dataset Effectiveness in Vehicle Trajectory Prediction

In this section, we evaluate the effectiveness of WOMD-Reasoning in a real-world driving task - vehicle trajectory prediction. Our experiments indicate that with our WOMD-Reasoning dataset involved, the metrics of vehicle trajectory predictions are significantly improved.

### 6.1. Language-assisted Trajectory Prediction Experiment Setup

We adapt the Multipath++ (Varadarajan et al., 2021) trajectory prediction model to accommodate the language input. The adaption mainly contains two steps:

**Language Acquisition and Encoding** The language we use comes from Motion-LLaVA's output. We run Motion-

*Table 4.* Benchmarks of fine-tuned multi-modal models with language metrics.

| Questions | Model | ROUGE (↑) | BLEU (↑) | METEOR (↑) | CIDEr (↑) | SPICE (↑) | GPT Score (↑) |
|---|---|---|---|---|---|---|---|
| All | Shuffled Ground-truths | 0.481 | 0.311 | 0.263 | 1.68 | 0.375 | 1.48 |
| | Non-Fine-tuned LLaVA (Liu et al., 2024) | 0.512 | 0.211 | 0.275 | 1.36 | 0.455 | 2.31 |
| | Fine-tuned LLaVA (Liu et al., 2024) | 0.779 | 0.581 | 0.439 | 5.51 | 0.735 | 6.88 |
| | Motion-LLaVA (**Ours**) | **0.792** | **0.616** | **0.449** | **5.69** | **0.744** | **7.02** |
| Facts Only | Shuffled Ground-truths | 0.528 | 0.349 | 0.273 | 2.26 | 0.401 | 2.06 |
| | Non-Fine-tuned LLaVA (Liu et al., 2024) | 0.561 | 0.322 | 0.326 | 1.50 | 0.483 | 2.46 |
| | Fine-tuned LLaVA (Liu et al., 2024) | 0.821 | 0.702 | 0.495 | 6.05 | 0.783 | 6.91 |
| | Motion-LLaVA (**Ours**) | **0.840** | **0.736** | **0.516** | **6.35** | **0.794** | **7.09** |
| Reasonings Only | Shuffled Ground-truths | 0.479 | 0.326 | 0.281 | 1.81 | 0.462 | 1.87 |
| | Non-Fine-tuned LLaVA (Liu et al., 2024) | 0.262 | 0.097 | 0.201 | 0.06 | 0.283 | 0.91 |
| | Fine-tuned LLaVA (Liu et al., 2024) | 0.596 | 0.452 | 0.357 | 2.26 | 0.557 | 6.14 |
| | Motion-LLaVA (**Ours**) | **0.614** | **0.474** | **0.366** | **2.52** | **0.571** | **6.76** |

LLaVA on the cases covered by WOMD-Reasoning dataset in the training and validation set of WOMD. For each scenario included, Motion-LLaVA would provide a comprehensive set of Q&As, where we pick the interaction Q&As for this experiment. Each of these Q&As includes the interaction info between a specific agent and the ego agent. We pick the answer parts of these Q&As to use. These answers are then fed into a T5 encoder (Raffel et al., 2023) followed by a few MLP layers to be encoded. In the end, for each scenario included, we have a set of language embeddings, each contains info on the interaction between the ego agent and one specific agent.

**Introducing Languages into Multipath++** Our next step is to introduce these encoded languages into each agent's feature. For Multipath++, we use MPA (Konev, 2022), an open-source Multipath++ implementation, as our code base. Our language introduction module is inserted right after their "Other Agents History Encoder", and before their "MCG (multi-context gating) encoder". Our module is a cross-attention block, which takes each agent's history encoding as queries, letting them cross-attend to their corresponding language embeddings containing info about the interaction between the ego agent and that specific agent. In this way, the interaction information in Motion-LLaVA is introduced into the features of each agent.

### 6.2. Enhancing Trajectory Prediction with Motion-LLaVA Outputs

*Table 6.* Benefits of Motion-LLaVA Outputs for Prediction Task.

| Model | $minFDE_6$ | $MR_6$ |
|---|---|---|
| Multipath++ (Konev, 2022) | 1.27 | 12.59 |
| + Motion-LLaVA Outputs | **1.18** | **11.69** |
| Δ | -7.35% | -7.10% |

As shown in Table 6, we observe significant metric improvements with language involved, which strongly supports WOMD-Reasoning's ability to help downstream tasks like predictions. It can also help to understand the driving behaviors, and to improve explainability of vehicle trajec-

tory prediction models.

## 7. Conclusion

We first build WOMD-Reasoning, a language-centered multi-modal dataset upon WOMD, focusing on analyzing and reasoning interactions. Specifically, our WOMD-Reasoning emphasizes interactions induced by traffic rules and human intentions, such as interactions governed by traffic lights and signs, and interactions like overtaking and following, which are both rarely covered in previous datasets. Besides, WOMD-Reasoning provides 3 million Q&A pairs, which makes it the largest real-world language dataset in autonomous driving to the best of our knowledge, covering a range of topics from scene descriptions, prediction, to planning. To demonstrate the applications of WOMD-Reasoning, we propose Motion-LLaVA, which fine-tunes a multi-modal model to accurately answer questions covering both driving scenario descriptions and interaction reasonings. Motion-LLaVA realizes the function of interaction prediction and traffic rule-compliant planning, taking advantages of the WOMD-Reasoning and the text-based knowledge intrinsically built in language models. Motion-LLaVA proves the effectiveness of WOMD-Reasoning in assisting real-world driving through supporting various language-involved downstream tasks.

## Acknowledgements

This work was supported in part by Berkeley DeepDrive, NSF Grant. 2235013, NSF Grant. 2235012, and the Microsoft Accelerate Foundation Models Research Initiative.

## Impact Statement

Our work is of broad interest to the natural language processing (NLP) and autonomous driving communities. The proposed approach has no new ethical or social issues on its own, except those inherited from NLP or autonomous driving.

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

# A. Appendix

In this appendix, we first show the building process of WOMD-Reasoning in detail in A.1. Specifically, we show the construction of the interpreted language of the motion data with an example (A.1.1), and we revisit the detailed methods utilizing GPT to build the language dataset as well as providing the full set of prompts (A.1.2). Some examples of the vision modal of WOMD-Reasoning provided by simulations are presented in A.2.

Then, for Motion-LLaVA, whose language output enhances trajectory prediction as shown in Section 6, we provide further implementation details. Specifically, in A.3, we describe how agent information from driving scenarios is incorporated into the input of Motion-LLaVA. The training hyperparameters used for the downstream trajectory prediction module are presented in A.4. The prompt used to obtain the GPT Score for evaluation is provided in A.5. We further provide a qualitative comparison in A.6 of the benchmarked multi-modal models listed in Table 4, and introduce in A.7 additional multi-modal models not included in the table. Examples of aggregated context (A.8) and factual description context (A.9) used in Chain-of-Thought interaction reasoning are also provided for reference. An ablation study on the fine-tuning strategy used in Motion-LLaVA is included in A.10, along with an analysis of the data efficiency of WOMD-Reasoning in helping Motion-LLaVA learn driving-related reasoning in A.11.

## A.1. Details in Building WOMD-Reasoning

In this appendix, we show the details of how we build WOMD-Reasoning dataset, including detailed explanations on each step we take, the example of the translated motion scenarios, and the GPT prompts. The full version of codes and prompts used herein can be found in the supplemental materials.

### A.1.1. AUTOMATIC TRANSLATION OF DRIVING SCENARIOS INTO LANGUAGE DESCRIPTIONS

ChatGPT cannot effectively process raw data from motion datasets, as it is mainly trained on textual information rather than motion data. Therefore, we first translate the motion information into textual descriptions by a rule-based program. Accurate motion translation is critical to the quality of the dataset, as it serves as the only input observable to GPT. Ensuring thorough and well-structured translations is essential for GPT to perform accurate interaction analysis and reasoning.

To ensure enough information for analyzing interactions involving the ego agent, we translate the motion data into an ego-centric description of the traffic scenario. Specifically, before describing the ego agent status, we first narrate the map environment around the ego agent, including information about related intersections, lanes, stop signs, crosswalks, speed bumps, etc. The above information provides a comprehensive spatial guidance that facilitates describing the motion of either ego or other agents accurately. Then, we describe the status of the ego agent in the scenario, including its motion status (e.g., the velocity, acceleration, etc.) and the positional status (e.g., the lane it occupies and its position related to the intersection center). Furthermore, the traffic light and signal information related to it are summarized in the description. Besides the ego agent, the depiction of other agents, including other vehicles, bicycles, and pedestrians, is also crucial for GPT to analyze the interactions. Therefore, we then describe every other agent in the scenario in the same way as we describe the ego agent. We additionally include the positional relation between the other agent and the ego one to support the identification of every agent's position.

Utilizing the methods above, we can translate motion datasets into languages. Due to the limitations of computational resources of GPT, we only choose to translate those scenarios that have `objects_of_interest` labels in WOMD, which indicates confirmed significant interactions happening in the scenario. We build two subsets of WOMD-Reasoning with the same setting: the training set is built on the training set of WOMD, while the validation set is built on the interactive validation set of WOMD. In total, we translate 63k scenarios into language. An example of the translation can be found in Table 7.

### A.1.2. GPT-BASED INTERACTION ANALYSIS

With the language translation of WOMD as input, we further design a set of prompts to utilize ChatGPT-4 (OpenAI et al., 2024) to build the Q&A dataset, taking advantage of its abilities in analyzing interactions. The prompts come with 4 main parts. **(1) System Prompt**, which lets the GPT know its role and the format of its input, i.e. the language translation. **(2) Responsibility Prompt**, which describes the responsibilities of the GPT, including the questions it should ask and answer, as well as the output format it should use. This will be talked about in detail in the next paragraph. **(3) Rules Prompt**, which contains global rules to guide the output and the analysis. **(4) In-context Prompt**, which provides a few human-written

**Descriptions on map environment:**
The ego agent is heading towards intersection. The intersection center is 18.0 meters in front of the ego agent, and is 17.0 meters on the left of the ego agent. The intersection is a 4 way intersection.
**Descriptions on ego agent:**
The ego agent is on the 1 lane from the right, out of 3 lanes. Its current speed is 8 m/s. It is accelerating. Traffic Light for the ego agent is red. The ego agent is approaching a crosswalk 3 meters ahead.
**Descriptions on surrounding (other) agents:**
Surrounding agent # 0 is a vehicle. It is 23 meters in front of the ego agent, and is 17 meters on the left of the ego agent. It is heading right of the ego agent. Its current speed is 6 m/s. It is accelerating. It is in the intersection. It is 29 meters away from the ego agent. It is approaching a crosswalk 3 meters ahead.
Surrounding agent # 2 is a vehicle. It is 11 meters on the right of the ego agent, and is 0 meters behind the ego agent. It is heading left of the ego agent. It is not moving. It is on the same side of the intersection as the ego agent. It is 34 meters away from the intersection center.
. . .
**Descriptions on agents after 3 seconds:**
The following is the description of the ego and surrounding agents after 3.0 seconds:
Surrounding agent # 0 will be 8 meters in front of the ego agent, and will be 5 meters on the left of the ego agent. It will be heading right of the ego agent. Its speed will be 7 m/s. It will be departing from the intersection. Looking from the agent's current angle, it will be on the same side of the intersection. It will be 18 meters away from the intersection center.
The ego agent will be 13 meters in front of the current place, and will be 2 meters on the right of the current place. It will be heading in the same direction as the current moment. Its speed will be 3 m/s. It will be departing from the intersection. Looking from the agent's current angle, it will be on the same side of the intersection. It will be 20 meters away from the intersection center.
Surrounding agent # 2 will be 13 meters on the right of the ego agent, and will be 9 meters behind the ego agent. It will be heading left of the ego agent. It will not be moving. Looking from the agent's current angle, it will be on the same side of the intersection. It will be 34 meters away from the intersection center.

. . .

*Table 7.* An example of the language translation of motion data.

examples of input-output pairs to perform in-context learning. An example of each part of the prompts can be found in Table 8, 9, 10, and 11 respectively.

The questions in the responsibility part of the prompt decide what would be present in the generated language Q&A dataset. To provide a comprehensive language dataset, and to present interaction analysis as accurately as possible, we organize the questions into a chain to enable a well-designed Chain-of-Thought (Wei et al., 2022; Chu et al., 2023): First, we ask questions about the map environment, the ego and other agents' motion. These questions would let the GPT provide Q&As useful for fine-tuning scene description agents while enhancing the GPT's understanding of the input information at the same time. Then, we ask questions about the interactions that occur in the observable future period. To include traffic rule-induced and intention-induced interactions, we ask the GPT to think about the interaction between each pair of agents by answering a sequence of questions: **Q1:** Are the two agents vehicles, and are they close to each other with traffic rules or signals governing their movements? If the answer is yes, we request that GPT analyze their yielding relations according to the traffic rules. Otherwise, we ask **Q2:** Are the two agents vehicle and pedestrian respectively with intention conflicts? If so, the vehicle generally should yield to the pedestrian for safety. If both questions do not help find the interaction relations, we come to **Q3:** Does the scene show patterns of certain intention-induced interaction? We provide descriptions of a few common interactions like overtaking, following, etc. GPT can choose the best fit from these patterns to try to cover more interaction types. Based on this pipeline, we can provide interaction analysis with rich information on traffic rule-induced and human intention-induced interactions. After that, we can finally wrap up the interactions between the ego and all other agents to provide the intention of the ego agent.

### A.2. Examples of Simulated Vision Modality

Figure 8 provides some whole examples of the simulated scenarios generated by MetaDrive (Li et al., 2022) simulator using ScenarioNet (Li et al., 2023) traffic scenario simulation platform.

**System prompt for GPT:**
You are an AI assistant to analyze real-world driving scenes.
You are provided with the detailed information (formated as: [start of the input] the detailed information [end of the input]) of the real-world driving scenario, which primarily includes information about the ego vehicle (denoted as 'Ego Agent'), other vehicles, cyclists or pedestrians (denoted as 'Surrounding agent #n'), intersections and road lanes. Note that the input includes the current situation as well as the situation in a future moment for your reference.
Unless you are told to guess, only provide information you are certain according to the input. The input is complete. Any agents or other things not mentioned in the input does NOT exist, and you should NOT guess what would happen if they exist.

*Table 8.* System prompt for GPT.

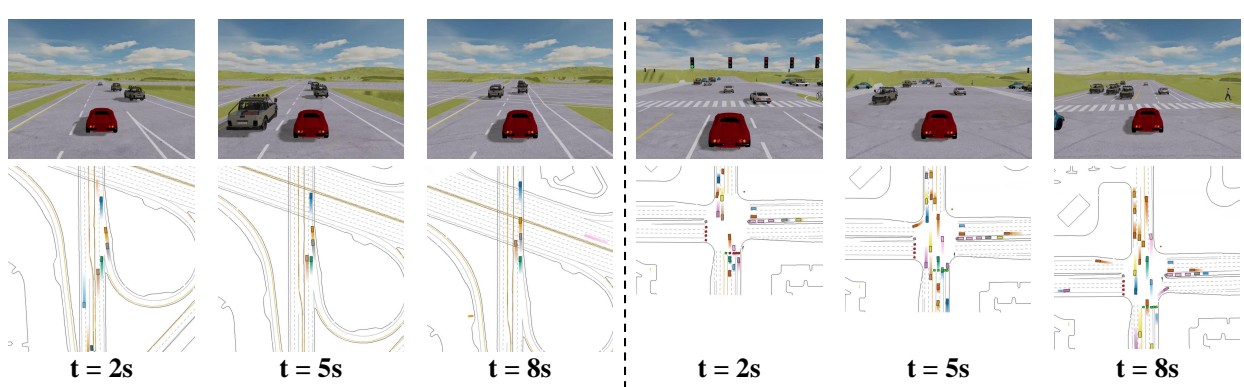

| **t = 2s** | **t = 5s** | **t = 8s** | **t = 2s** | **t = 5s** | **t = 8s** |

*Figure 8.* More visual examples of WOMD-Reasoning through simulations.

### A.3. Processing Agent IDs for Motion-LLaVA

Careful processing of agent IDs is essential for enabling Motion-LLaVA to accurately distinguish between different agents. In our approach, these IDs are used purely as textual identifiers within the prompts, allowing the LLM to differentiate agents through natural language understanding. The prompt format is:

```
Ego agent:  <motion>\n Agent #0:  <motion>\n...\n Now, please answer:  {Question}
```

, where <motion> denotes the encoded motion data from each agent's perspective. The IDs are assigned by range: #0–#99 for vehicles, #100–#199 for bicycles, and #200–#299 for pedestrians. We remap WOMD agent IDs to these local ranges to prevent downstream models from overfitting to specific agents' behaviors. Importantly, these agent IDs do not encode any additional information beyond their role as identifiers.

### A.4. Hyperparameters for Multipath++

In the experiment using languages from Motion-LLaVA outputs to assist vehicle trajectory predictions, we use a batch size of 128 and a learning rate of $1e-4$ for Multipath++. As language is not available for every agent, we use `"No captions available for this agent."` for those agents without language inputs. The $t5-v1\_1-xxl$ model is used as the language encoder.

### A.5. Prompts for evaluating GPT Score

We employ the OpenAI GPT-4-turbo model (OpenAI et al., 2024) to evaluate the performance of all models benchmarked on the WOMD-Reasoning. The GPT Score prompt is presented in Table 12. Notably, we design our GPT Score prompt using a similar template to the one from (Chen et al., 2023).

### A.6. Qualitative Comparison of Benchmarked Multi-modal Models on Answering Questions in WOMD-Reasoning

In this appendix, we present examples of answers to questions in WOMD-Reasoning generated by the benchmarked multi-modal models listed in Table 4. The selected scenario corresponds to Figure 7, which already showcases answers from

Motion-LLaVA. Additionally, we provide results for both the Non-Fine-tuned (i.e., off-the-shelf) and Fine-tuned LLaVA models, both using BEV plots as vision input.

For the Non-Fine-tuned LLaVA, we perform prompt engineering to make the answers coherent. However, the low GPT scores indicate that it remains challenging for LLaVA to directly comprehend the scenario:

- **Q:** Can you describe the type of intersection present in the current driving scenario?
  **A:** The type of intersection present in the current driving scenario is a four-way intersection.
  **GPT Score:** 0. Incorrectly identifies the type of intersection.

- **Q:** What is the ego agent's current motion status?
  **A:** The ego agent is currently stopped at the intersection.
  **GPT Score:** 0. Incorrectly describes the motion status of the ego agent.

- **Q:** Where is surrounding agent #5 in relation to the ego agent?
  **A:** Surrounding agent #5 is located to the right of the ego agent.
  **GPT Score:** 0. Incorrect location of surrounding agent #5 in relation to the ego agent.

- **Q:** What kind of interaction will occur between the ego agent and surrounding agent #5?
  **A:** The ego agent will pass by the surrounding agent #5 at the intersection.
  **GPT Score:** 0. Incorrect prediction of the interaction between the ego agent and surrounding agent #5.

Quantitatively, the Non-Fine-tuned LLaVA achieves a remarkably low GPT Score of 2.31 across all questions in the validation set, demonstrating difficulty in understanding the driving scenarios based solely on the prior knowledge of the Vision-Language Model (VLM). For comparison, we evaluate the performance of the Fine-tuned LLaVA with vision input under the same scenario. The fine-tuned model achieves much more reasonable results, as illustrated below:

- **Q:** Can you describe the type of intersection present in the current driving scenario?
  **A:** The intersection is a 3 way intersection.
  **GPT Score:** 10. The student's answer matches the correct answer exactly.

- **Q:** What is the ego agent's current motion status?
  **A:** The ego agent is accelerating.
  **GPT Score:** 10. The student's answer matches the correct answer exactly.

- **Q:** Where is surrounding agent #5 in relation to the ego agent?
  **A:** It is 14 meters on the right and 10 meters in front of the ego agent.
  **GPT Score:** 0. The student's answer incorrectly describes the position of surrounding agent #5.

- **Q:** What kind of interaction will occur between the ego agent and surrounding agent #5?
  **A:** Surrounding agent #5 will follow the ego agent as they are both heading towards the intersection and are on the same side.
  **GPT Score:** 5. The student's answer is partially correct but does not specify that agent #5 is behind the ego agent.

The results indicate that fine-tuning significantly improves LLaVA's ability to interpret driving scenarios. This improvement can be attributed to the inclusion of our WOMD-Reasoning dataset. However, despite these enhancements, notable under-performance persists in questions related to **Other Agent Status** and **Interaction & Intention**, especially when compared to the results in Figure 7, which are generated using Motion-LLaVA. We attribute this limitation to potential information loss during the conversion of vectorized motion data into a BEV plot with a scalar bar.

### A.7. Additional Baseline Language Models on WOMD-Reasoning

Our main results in Table 4 primarily evaluate different variants of LLaVA (Liu et al., 2024) to demonstrate the effectiveness of WOMD-Reasoning. For clarity, we explain the categories of questions assessed. The questions are grouped into two main types: factual questions and interaction reasoning questions.

- **Factual questions:**

- **Environment-related:** Questions about the scenario environment, such as the existence and type of intersections, the number of lanes on the ego vehicle's side, and the presence or location of crosswalks and stop signs.
- **Ego agent-related:** Questions regarding the ego vehicle's properties, including speed, motion status, lane position, and direction.
- **Surrounding agents-related:** Questions about other agents in the scenario, covering their type, speed, motion status, and position relative to the ego vehicle and the intersection.

- **Interaction reasoning questions:**

  - Questions about the interactions that will occur between each surrounding agent and the ego agent.
  - Questions about the overall intention of the ego agent in the given scenario.

We further evaluate additional language models (LMs) to confirm the effectiveness of WOMD-Reasoning. Specifically, we assess the answer quality of several baseline models, including LLaMA-Adapter v2.1 (Gao et al., 2023), VITA-1.5 (Fu et al., 2025), and Qwen2.5-VL (Bai et al., 2025) (all 7B versions), both before and after fine-tuning on WOMD-Reasoning. The results are summarized in Table 13.

We observe that without fine-tuning, all models can hardly answer driving-related questions, supporting the motivation of building WOMD-Reasoning. We then fine-tune LLaMA-Adapter on WOMD-Reasoning, which also significantly benefit from the fine-tuning. Besides, we find that fine-tuned our Motion-LLaVA works better than fine-tuned LLaVA or LLaMA-adapter, proving its well-designed structure in utilizing WOMD-Reasoning information.

### A.8. Example Aggregated Context for Chain-of-Thought Reasoning

We present an example of an **aggregated** context, derived from **the answers generated by Motion-LLaVA** for factual questions, which is used by Motion-LLaVA to infer answers to reasoning-based questions. This context corresponds to the **CoT** configuration in Table 14:

The ego agent is turning left. Its current speed is 5 m/s. It is accelerating.

Surrounding agent #7 is a vehicle. It is 26 meters on the left of the ego agent and 8 meters in front. It is heading right of the ego agent. Its current speed is 3 m/s. It is decelerating.

The blue and orange sections contain answers to questions appearing in WOMD-Reasoning: blue addresses the Road Environment and Ego Agent Status, while orange covers Other Agent Status.

To generate this context, we prompt the Motion-LLaVA model with relevant questions and then concatenate the answers to create a complete context.

### A.9. Example Factual Description Context for Chain-of-Thought Reasoning

We provide an example of a factual description context, generated using our motion data translation program as part of constructing WOMD-Reasoning, as described in Appendix A.1.1. This context is used by Motion-LLaVA to infer answers to reasoning-based questions and corresponds to the **CoT w/ Facts Contexts** configuration in Table 14. Notably, these contexts exclude future information compared to the translated motion data, thereby avoiding any information leakage:

The ego agent is in intersection. The intersection center is 6.0 meters in front of the ego agent, and is 0.0 meters on the left of the ego agent. The intersection is a 4 way intersection. It is turning left. It is exiting the intersection. Its current speed is 3 m/s. It is accelerating. Traffic Light for the ego agent is green. The ego agent is approaching a crosswalk 3 meters ahead.

Surrounding agent #7 is a vehicle. It is 26 meters on the left of the ego agent, and is 8 meters in front of the ego agent. It is heading right of the ego agent. Its current speed is 4 m/s. It is decelerating. It is heading towards the intersection. It is on the left of the intersection. Traffic Light for it is red 14.5 meters ahead. It is 26 meters away from the intersection center. It is approaching a crosswalk 4 meters ahead.

Similar to the aggregated context in Appendix A.8, the blue and orange sections contain answers to questions appearing in WOMD-Reasoning: blue addresses the Road Environment and Ego Agent Status, while orange covers Other Agent Status.

### A.10. Ablations on Motion-LLaVA

**Chain-of-Thought Inference.** To address the hallucinated response problem (Gai et al., 2024; Tonmoy et al., 2024; Sima et al., 2023) in reasoning interactions, we use a Chain-of-Thought (CoT) approach to provide Motion-LLaVA with structured context for more accurate reasoning. Table 14 compares model performance with and without CoT on interaction and intention analysis. We see that with CoT, the model performs better in GPT Scores. We further show that if our motion data translation program A.1.1 is used during inference to provide contexts, i.e., with factual description context (an example can be found in Appendix A.9) and CoT, the performance can be significantly improved, proving the potential of the CoT inference strategy.

**Influence of Pre-training for the Motion Vector Encoder.** To establish a pre-training strategy for the motion vector encoder, we examine factual question answering across Motion-LLaVA with four pre-training configurations: no pre-training, autoencoder-style pre-training, motion-caption pre-training (Chen et al., 2023), and pre-training with a motion prediction task. All configurations shared the same encoder structure. In the motion-caption pre-training, we pre-train all but LLM weights in Motion-LLaVA, by penalizing errors in captioning the motion. For autoencoder-style pre-training and pre-training with a motion prediction task, we retain only the encoder after pre-training, to function as the motion vector encoder for Motion-LLaVA.

The performances of models with these pre-training settings are presented in Table 15. Generally, pre-training motion vector encoders can improve the response quality. Specifically, pre-training with motion prediction tasks outperforms the other two approaches. We attribute this to the close alignment between WOMD-Reasoning 's question categories and the information needed in motion prediction tasks. As a result, pre-training with the motion prediction task achieves the highest GPT Score and overall accuracy, especially in numerical precision reflected by $Acc_{1.0}$.

**Motion-LLaVA Fine-tuning Strategy.** The original LLaVA (Liu et al., 2024) training pipeline consists of two stages: a projector alignment stage (also known as pre-training) to align vision and language features, followed by a fine-tuning stage to enhance instruction-following capabilities. Throughout this process, the vision encoder remains frozen. Since our Motion-LLaVA builds on the architecture of the original LLaVA (Liu et al., 2024), it seems reasonable to adopt a similar training approach for the WOMD-Reasoning task.

However, based on extensive experiments with fine-tuning strategies in Table 16, we find that incorporating projector alignment stage or keeping the motion vector encoder frozen throughout, which are practices employed in LLaVA (Liu et al., 2024), resulted in degraded performance on WOMD-Reasoning. Consequently, our Motion-LLaVA adopts a single-stage fine-tuning approach, where the motion vector encoder is unfrozen.

### A.11. WOMD-Reasoning Dataset Efficiency Analysis

To assess the data efficiency of WOMD-Reasoning, we conduct scaling studies to analyze how model performance improves as the used dataset's size increases, from 0.5M to 1M and up to 3M Q&A pairs (full dataset).

In Table 17, we find that with more WOMD-Reasoning data involved, the fine-tuned model first gains the ability to answer factual questions, which hits a good GPT score with only 0.5M data. As the dataset grows, the model gradually learns to answer interaction reasoning questions. These justify the size of WOMD-Reasoning, as well as its data efficiency in helping models to answer hierarchical driving-related questions.

**Responsibility Prompt for GPT:**
You are responsible for the following responsibilities:
<First>, print "[Env QA]". Generate Q&A on all the scenario description inputs about the scenario environment. The Q&A must be in the following format: "[Q][Question content] [A][Answer content]" The Q&A should cover ALL information about the CURRENT moment in the input description. No information in the provided future moment should be used. Your questions should include all the questions in the following categories except the answer is not mentioned in the input: (1) About the intersection: its existence? Its type? (2) About the lanes: How many lanes on ego's side? (3) About stop signs: its existence? where is it? which direction is it for?
. . .

---

<Second>, print "[Ego QA]". Generate Q&A on all the scenario description inputs about the the ego agent. The Q&A must be in the following format: "[Q][Question content] [A][Answer content]" The Q&A should cover ALL information about the CURRENT moment in the input description. No information in the provided future moment should be used. Your questions should include all the questions in the following categories: (1) About its motion: its speed? its motion status? (2) About its position: its position? its lane? (3) About its direction: its facing direction? its turning direction?
. . .

---

<Third>, print "[Sur QA]" and generate Q&A on all the scenario description inputs about surrounding agents. The Q&A must be in the following format: "[Q][Question content] [A][Answer content]" The Q&A should cover ALL information about the CURRENT moment in the input description. No information in the provided future moment should be used. For each surrounding agent, your questions should include all the questions in the following categories except the answer is not mentioned in the input: (1) About itself: its type? (2) About its motion: its speed? its motion status? (3) About its position: its position to the ego agent? its position to the intersection? its lane?
. . .

---

<Fourth>, start a new paragraph and print "[Int QA]". For each surrounding agent, start a new paragraph and provide one Q&A for the interactions between it and the ego agent. The Q&A must be in the following format: "[Q][Question content] [A][Answer content]" Here [Question content] should have the meaning "What interactions will happen between surrounding agent #n and the ego agent?" without any additional information like "considering what" or "given what", but you must use diverse ways to put it. (i.e. ask in different ways for different surrounding agents) When providing [Answer content], you should use the descriptions for both the current moment and the future moment, and analyze based on them. Pay attention to the special situations where you should consider that there is an interaction:
. . .

---

<Fifth>, for the ego agent, start a new paragraph, print "[Intention]", and provide one Q&A for the intention of the ego agent. The Q&A must be in the following format: "[Q][Question content] [A][Answer content]" Here [Question content] should have the meaning "What will be the intention of the ego agent?", but you must use diverse ways to put it. When providing [Answer content], and think about following questions in the time period from now to the provided future moment based on given description: (1) What is the ego agent's intention if no other agents exist? (2) If applicable, what actions does the ego agent take to respond to traffic lights, stop signs, crosswalks, speed bumps and traffic rules? If some traffic controls are not applicable, do not output on that. (3) In each interaction involving the ego agent, what actions do the ego and surrounding agent take to respond to each other? If one agent does not have interaction with the ego agent, exclude it in your answer.
. . .

---

Remember to use diverse ways to put the [Answer content]. Do not include any numbers regarding speeds or positions in your [Answer content].

*Table 9.* Responsibility prompt for GPT.

**Global rules prompt for GPT:**

Here are some global rules that your need to follow:

1. Please provide the summary and analysis of the driving scenario according to your responsibilities.

2. Never use bullet points. Be concise and compact.

3. Give natural language, no parenthesis.

4. Give the results in the order of the responsibilities. Always start a new paragraph for each responsibility, but one responsibility can take several paragraphs. Do not combine multiple responsibilities together. Finish one responsibility before starting another.

5. Do not show the intersection number, lane number or the ego agent's number. The surrounding agents' numbers need to be shown.

6. Output in full. Never omit anything regardless of any reason. You are allowed to give very long outputs.

7. Remember that all the agents and traffic controls in the scene have been provided, so if any agents or traffic controls are not mentioned in the scene, they do not exist. You don't need to consider them or give any output about them. Therefore, do not output words like "not mentioned" because this is inaccurate.

8. The traffic light info is for the current moment only. The future traffic light info is unknown.

9. Any action caused by traffic controls like lights, stop signs or crosswalks may also constitute yielding interactions. Traffic controls are only an indication of right-of-way, but any actions that are caused by the traffic controls are still considered interactions. For example, if a car is waiting for the red light or a stop sign while someone other cross the intersection with green light, the car is still considered yielding to the other car.

10. When analyzing the future, talk about the period between the current and future moment provided, not the two moments.

*Table 10.* Rule prompt for GPT.

**Context prompt for GPT:**
Here is one given example that you can refer to to help you better understand you responsibility <Fourth> and <Fifth>: For the input case of:
[start of input]
The ego agent is in intersection. The intersection center is 10.0 meters in front of the ego agent, and is 1.0 meters on the left of the ego agent. The intersection is a 4-way intersection.
It is going straight. It is exiting the intersection. It 's current speed is 6 m/s. It is accelerating. The ego agent is approaching a stop sign 1 meters ahead. There are 4 stop signs in the intersection. The ego agent is approaching a crosswalk 1 meters ahead. Surrounding agent # 0 is a vehicle. It is 4 meters on the left of the ego agent, and is 2 meters behind the ego agent. It is heading the opposite direction as the ego agent. Its current speed is 9 m/s. It is accelerating. It is departing from the intersection. It is on the same side of the intersection as the ego agent. It is 12 meters away from the intersection center.
· · ·
[end of input]

you may give the following analysis:
[Int QA]
[Q] What kind of interaction can be expected between the ego agent and surrounding agent #0 given their positions and directions?
[A] Surrounding agent #0 will have no interaction with the ego agent as their intended paths have no conflicts.
· · ·
[End Int QA]

[Intention]
[Q] What will the ego agent aim to do in the upcoming moments?
[A] The ego agent intends to continue exiting the intersection. There is a stop sign on its side but it has already stopped and started afterwards. Surrounding agent #1 will yield to it so the ego agent does not need to respond. Surrounding agent #3 will follow it but the ego agent does not need to respond as well. Therefore, the ego agent will continue in the intersection and finish its left turn.
[End Intention]

Here is another given example that you can refer to to help you better understand you responsibility <Fourth> and <Fifth>:
[start of input]
The ego agent is heading towards intersection. The intersection center is 28.0 meters in front of the ego agent, and is 3.0 meters on the left of the ego agent. The intersection is a 3 way intersection.
The ego agent is on the 2 lane from the left, out of 3 lanes. It 's current speed is 0 m/s. It is decelerating. Traffic Light for the ego agent is red 9.6 meters ahead. The ego agent is approaching a crosswalk 3 meters ahead.
Surrounding agent # 0 is a vehicle. It is 29 meters in front of the ego agent, and is 4 meters on the right of the ego agent. It is heading left of the ego agent. Its current speed is 6 m/s. It is accelerating. It is in the intersection. It is 29 meters away from the ego agent. The ego agent is at a speed bump.
· · · [end of input]

you may give the following analysis:
[Int QA]
[Q] What kind of interaction can be expected between the ego agent and surrounding agent #0?
[A] The ego agent will yield to surrounding agent #0 because they are both approaching the intersection and the traffic light is red for the ego agent, indicating that the ego agent must stop.
[End Int QA]

[Intention]
[Q] What will the ego agent aim to do in the upcoming moments?
[A] The ego agent intends to enter the intersection, but it has to stop at the red traffic light and wait for it to change to green before proceeding through the intersection. It will also yield to surrounding agents #0 who is already in the intersection. Therefore, the ego agent will wait until the lights turn green and the surrounding agent #0 passes before entering the intersection. [End Intention]

· · ·

*Table 11.* In-context prompt for GPT.

We would like to request your feedback on the performance of an AI assistant in response to the user question displayed above. The user asks the question on observing a driving scenario. The correct answer is included for your reference.
### Scoring rules:

- Your scores for each answer should be between 0 (worst) and 10 (best).

- If the answer is mostly incorrect but includes important relevant information, such as identifying correct agents or actions, give a score of less than 3.

- Give intermediate scores for partially correct answers or when the numbers are close. Only give a score of 10 for flawless answers.

- Think critically and carefully. Most importantly, check the answer for factual correctness, but reward partial credit where applicable.

### Grading process For each of the xxx questions, provide a one line assessment of the student's answer in the format:
"' n. <short one sentence assessment>. Score: <score 0-10>. "'
A few examples of what a good assessment might look like:

1. Acknowledged question unrelated to driving and attempted to answer. Score: 10.

2. Doesn't answer the question but has all information necessary. Score: 3.

3. Incorrectly stated there are no vehicles around even though there is one. Score: 0.

4. Unable to answer question given information available. Score: 10.

5. Give 12 m/s or m or percentage for the correct answer of 13 (within 10% deviation). Score: 9.

6. Give 15 m/s or m or percentage for the correct answer of 13 (within 20% deviation). Score: 7.

7. Give 18 m/s or m or percentage for the correct answer of 13 (within 40% deviation). Score: 5.

8. Give 2 m/s or m or percentage for the correct answer of 13. Score: 0.

*Table 12.* Prompt for GPT Score.

*Table 13.* Additional language model baselines evaluated on WOMD-Reasoning. We report the answer quality of several vanilla and WOMD-Reasoning-fine-tuned models, including LLaMA-Adapter, ViTA, and Qwen2.5-VL (all 7B versions), to further validate the effectiveness of WOMD-Reasoning.

| Model | Fine-tuned on WOMD-Reasoning | ROUGE (↑) | BLEU (↑) | METEOR (↑) | CIDEr (↑) | SPICE (↑) | GPT Score (↑) |
|---|---|---|---|---|---|---|---|
| LLaVA | ✗ | 0.512 | 0.211 | 0.275 | 1.36 | 0.455 | 2.31 |
| LLaMA-Adapter v2.1 | ✗ | 0.413 | 0.174 | 0.235 | 0.91 | 0.372 | 1.62 |
| ViTA-1.5 | ✗ | 0.278 | 0.121 | 0.190 | 0.40 | 0.227 | 1.77 |
| Qwen2.5-VL | ✗ | 0.384 | 0.156 | 0.247 | 0.62 | 0.379 | 2.36 |
| LLaVA | ✓ | 0.779 | 0.581 | 0.439 | 5.51 | 0.735 | 6.88 |
| LLaMA-Adapter v2.1 | ✓ | 0.722 | 0.470 | 0.375 | 4.72 | 0.691 | 5.14 |
| Motion-LLaVA (Ours) | ✓ | **0.792** | **0.616** | **0.449** | **5.69** | **0.744** | **7.02** |

*Table 14.* Effects of CoT in answering reasoning-related questions.

| Reasoning Inference Method | ROUGE (↑) | BLEU (↑) | METEOR (↑) | CIDEr (↑) | SPICE (↑) | GPT Score (↑) |
|---|---|---|---|---|---|---|
| Direct | 0.616 | 0.474 | 0.365 | 2.54 | 0.578 | 6.59 |
| CoT | 0.614 | 0.474 | 0.366 | 2.52 | 0.571 | 6.76 |
| CoT w/ Facts Contexts | **0.664** | **0.542** | **0.403** | **3.10** | **0.631** | **7.64** |

*Table 15.* Benefits of using pre-trained motion vector encoder for fine-tuning multi-modal models on WOMD-Reasoning for factual Q&As.

| Motion Vector Encoder Pre-training | ROUGE (↑) | BLEU (↑) | METEOR (↑) | CIDEr (↑) | SPICE (↑) | $Acc_{1.0}$ (↑) | Median Error(↓) | GPT Score (↑) |
|---|---|---|---|---|---|---|---|---|
| None | 0.814 | 0.684 | 0.491 | 5.95 | 0.774 | 30.0% | 2.00 | 6.79 |
| Motion Caption | 0.824 | 0.711 | 0.498 | 6.05 | 0.775 | 42.6% | 2.24 | 6.97 |
| AutoEncoder | 0.837 | 0.727 | **0.516** | **6.35** | **0.803** | 47.3% | 1.41 | 7.07 |
| Motion Prediction (Ours) | **0.840** | **0.736** | **0.516** | **6.35** | 0.794 | **56.2%** | **1.00** | **7.09** |

*Table 16.* Refinement of Motion-LLaVA training strategy.

| Multi-modal Fine-tuning Strategy | Projector Alignment | Encoder Freezing | ROUGE (↑) | BLEU (↑) | METEOR (↑) | CIDEr (↑) | SPICE (↑) | $Acc_{1.0}$ (↑) | Median (↓) | GPT Score (↑) |
|---|---|---|---|---|---|---|---|---|---|---|
| LLaVA | ✓ | ✓ | 0.821 | 0.678 | 0.484 | 5.82 | 0.771 | 2.3% | 12.00 | 6.17 |
| — Projector Alignment | ✗ | ✓ | 0.821 | 0.697 | 0.492 | 5.96 | 0.782 | 2.3% | 8.06 | 6.49 |
| — Encoder Freezing (**Ours**) | ✗ | ✗ | **0.840** | **0.736** | **0.516** | **6.35** | **0.794** | **56.2%** | **1.00** | **7.09** |

*Table 17.* Performance of Motion-LLaVA with different WOMD-Reasoning dataset sizes on factual and interaction reasoning questions. Results confirm the data efficiency of WOMD-Reasoning in improving driving-related reasoning.

| Dataset Size | Factual Questions | | | | | | Interaction Reasoning Questions | | | | | |
|---|---|---|---|---|---|---|---|---|---|---|---|---|
| | ROUGE | BLEU | METEOR | CIDEr | SPICE | GPT | ROUGE | BLEU | METEOR | CIDEr | SPICE | GPT |
| 0.5M | 0.806 | 0.683 | 0.477 | 5.77 | 0.760 | 6.69 | 0.562 | 0.410 | 0.339 | 1.94 | 0.513 | 5.67 |
| 1M | 0.818 | 0.701 | 0.491 | 5.90 | 0.773 | 6.74 | 0.589 | 0.447 | 0.354 | 2.23 | 0.547 | 6.36 |
| 3M | **0.840** | **0.736** | **0.516** | **6.35** | **0.794** | **7.09** | **0.614** | **0.474** | **0.366** | **2.52** | **0.571** | **6.76** |

