# OpenReview forum: "WOMD-Reasoning: A Large-Scale Dataset for Interaction Reasoning in Driving"
_ICML.cc/2025/Conference — ICML 2025 poster_

### Official Review · Reviewer_Qnzh · 2025-03-12

**Overall Recommendation:** 3

**Summary:**

The authors provided the new Q&A Reasoning dataset built on top of the famous Waymo Open Motion Dataset (WOMD-Reasoning) with the help of ChatGPT-4 and MetaDrive simulator for visualization purposes, and checked the baseline performance of Motion-LLaVA on top of it.

## update after rebuttal
Authors did a slight downstream testing and increased a number of Human-evaluated QA pairs, so I increased the overall score from WR to WA

**Claims And Evidence:**

Claims:
1. Providing of WOMD-Reasoning, the largest multi-modal dataset with 3 million Q&A pairs focused on interaction reasoning in driving.
2. Fine-tuning on WOMD-Reasoning, Motion-LLaVA provides detailed and insightful interaction predictions on various driving scenarios
3. Ablation study on Motion-LLaVA design

Evidence:
1. To be shared but there is no doubt that it is done, also refer to the Tables 1 and 2.
2. and 3. Using LLaVA + Multipath++ encoders (Figure 4), authors finetuned the solution to get the results shown in Tables 4 and 5 underlining the correct approach / architecture used; some ablations are shown in Tables 13 and 14 (Appendix)

**Essential References Not Discussed:**

NA

**Experimental Designs Or Analyses:**

* Quantitative analysis concentrates mostly on Language-related metrics - and very sparsely on Acc / Median Error. That's far from being useful
* In Section 5.4 "Validation of Motion-LLaVA" there is no details on what type of "facts" were measured (map elements, other agents states, ego state). Moreover, was Motion-LLaVA fine-tuned with motion prediction task or not?

**Methods And Evaluation Criteria:**

* One of the main drawbacks is the validity of the automatically generated by ChatGPT data - especially about intentions. Authors tried to assess the quality of auto labels by human evaluation in Section 4.3 - "4 people to judge 1610 Q&A". This is quite small in comparison to the size of the dataset (3 millions of Q&A), less than 1%.
* Moreover, the results of human assessment - e.g. 87.5% for Intentions - is quite low, especially for Planning, where each % is prone to be converted to some disastrous accident on the road
* There is no information provided on whether the incorporation of WOMD-Reasoning into any Autonomous Driving solution actually helps with any of real Autonomous Driving metrics (safety / comfort / progress), or even improves just the prediction power of common methods.

**Other Comments Or Suggestions:**

Typo:
* Line 149: "some studies x"

**Other Strengths And Weaknesses:**

Main items were emphasized in "Methods And Evaluation Criteria" and "Experimental Designs Or Analyses" sections

**Questions For Authors:**

NA

**Relation To Broader Scientific Literature:**

Two main directions: Autonomous Driving and Multi-modality Large Language Models. The paper is to combine these areas on the task of reasoning for AD.

**Theoretical Claims:**

No theory involved

---

> ### Author Rebuttal · Authors · 2025-04-01
>
> We appreciate the reviewer's insightful and constructive feedback. We address your comments and concerns below.
>
> > Q1. Human Assessment Scale.
>
> We are grateful for the reviewer's suggestion on providing stronger evidence of validations. To explore the validity of our assessment, we further enlarge our assessment to ~2.2k Q&As. With that being done, the total correct rate comes from 91.99% in the paper to 91.80%, which does not fluctuate much. This suggests that our sampling for assessment is valid and effective. Besides, we hope to confirm that our dataset improvement and human verifications are continuously ongoing. We will update the dataset and the validity when newer data become available.
>
> > Q2. Real-world risks of imperfect (intention) languages.
>
> We thank the reviewers for raising this question. We hope to address that, since language cannot thoroughly describe exact positions and velocities etc., language models are more commonly being used as an auxillary driving model to provide explainability and introduce high-level knowledge (like traffic rules in our case) or explainability. Therefore, its failing in infrequent cases would generally not cause accidents due to its auxillary nature. Furthermore, our intentions part serve as the summary of interactions between ego agent and all other agents, which is a very comprehensive task than other classes, therefore we believe that 87.5% is satisfying for an auto-labeled dataset. To further improve our dataset's quality, we will also update our dataset with human verifications to further ensure its reliability to 99+%.
>
> > Q3. Downstream impacts on real-world Autonomous Driving tasks.
>
> We highly appreciate this question. To justify WOMD-Reasoning dataset's ability in downstream tasks, we perform a trajectory prediction task using outputs of our Motion-LLaVA model fine-tuned upon WOMD-Reasoning.  Multipath++ is used as the trajectory prediction baseline model. As the interaction part is the most significant part of our dataset, we introduce the interaction part of language output into Multipath++, by cross-attending the T5 embedding of them with corresponding agents' features. Each experiment is runned with 3 seeds to ensure reliability. The results are shown in the following table
>
> | Model | minFDE_6 | MR_6 |
> |---|---|---|
> | Multipath++ | 1.27 | 12.59% |
> | Multipath++ with Interaction Language | **1.18** | **11.69%** |
> | relative &Delta; | -7.35% | -7.10% |
>
> We observe a significant performance enhancement using language outputs from motion-LLaVA fine-tuned upon WOMD-Reasoning. This strongly supports WOMD-Reasoning's ability to fine-tune LMs to help downstream tasks like predictions or explainabilities of driving behaviors. We will add final results of these experiments into our manuscript in our next version.
>
> > Q4. More quantitative analysis.
>
> We thank the reviewer for suggesting more downstream useful evaluations to prove WOMD-Reasoning's abilities. In Q3 we provide evidence proving the effectiveness of WOMD-Reasoning in trajectory prediction tasks. Since WOMD is a highly-interactive real-world driving dataset, prediction evaluations on WOMD offer evidences of the usefulness of WOMD-Reasoning on real-world applications.
>
> > Q5. Validation of Motion-LLaVA - "facts" measured.
>
> We thank the reviewers for requesting details on the definition of facts in Motion-LLaVA evaluations. Our Facts questions can be categorized into 3 types:
> 1) **Environment-related questions** focus on the scenario’s surroundings, covering details such as the presence and type of intersections, the number of lanes on the ego vehicle’s side, and the existence and location of crosswalks or stop signs.
> 2) **Ego agent-related questions** assess the ego vehicle's characteristics, including its speed, motion status, lane position, and directional attributes.
> 3) **Surrounding agents-related questions** focus on other agents in the scenario, covering their type, speed, motion status, and position relative to the ego vehicle and intersection.
>
> We hope this clarifications help, and we will include this in our next version.
>
> > Q6. Was Motion-LLaVA fine-tuned with motion prediction task or not?
>
> We are grateful for the question. While Motion-LLaVA is not fine-tuned with motion prediction task, we additionally use Motion-LLaVA outputs to illustrate the effectiveness of WOMD-Reasoning data in motion prediction tasks, the table is shown in Q3. We will add final results of these experiments into our manuscript in our next version.
>
> > Q7. Typo in Line 149
>
> We thank the reviewer for pointing this typo out. We will fix this in our next version.

---

> > ### Comment · Reviewer_Qnzh · 2025-04-02
> >
> > Thank authors for your careful investigation of the items I listed, esp. the downstream task impact.
> > I'd like to kindly ask for more details about:
> > * How this cross-attention module for Multipath++ was deployed (details on the architecture and training)
> > * Some examples on "intentions part serve as the summary of interactions between ego agent and all other agents" to see the real impact/importance of it

---

> > > ### Author Response · Authors · 2025-04-03
> > >
> > > We thank the reviewer for reading our response. Please see our answers to your questions below
> > >
> > > > Q1. How this cross-attention module for Multipath++ was deployed
> > >
> > > We appreciate the question on the details of introducing languages into Multipath++. This mainly contains 2 steps:
> > > 1) **Language Acquisition and Encoding**.
> > > The language we use comes from Motion-LLaVA's output. For each scenario, Motion-LLaVA would provide a comprehensive set of Q&As, where we pick the interaction Q&As for this experiment. Each of these Q&As include the interaction info between a specific agent and the ego agent (the one whose trajectory is to be predicted). We pick the answer parts of these Q&As to use. These answers are then fed into a T5 encoder followed by a few MLP layers to be encoded. In the end, for each scenario, we have a set of language embeddings, each contains info on the interaction between the ego agent and one specific agent.
> > > 2. **Introducing Languages into Multipath++**
> > > Our next step is to introduce these encoded languages into each agent's feature. For Multipath++, we use MPA [1], an open-source Multipath++ implementation, as our codebase, whose structure can be found in their Figure 1 (https://arxiv.org/pdf/2206.10041). Our language introduction module is inserted right after their "Other Agents History Encoder" (and before their "MCG (multi-context gating) encoder"). Our module is a cross-attention block, which takes each agent's history encoding as queries (shape = [agent_num, 1, feature_size]), letting them cross-attend to their corresponding language embeddings (shape = [agent_num, language_length, feature_size]) containing info about the interaction between the ego agent and that specific agent. In this way, the interaction information in Motion-LLaVA is introduced into the features of each agent.
> > >
> > > For experiment setups, due to limited computational resources, we use a subset of WOMD training set and WOMD validation-interactive set to perform the training and the evals of Multipath++, respectively. We choose the subsets using the same standard as the one we choose scenarios to build WOMD-Reasoning, where we pick scenes considered interactive by WOMD as flagged with the interactive agent pairs label in the `objects_of_interest` key, and the ego agent must be one side of the interactive pair. Eval metrics are minFDE_6 and MR_6 (Miss Rate).
> > >
> > > [1] MPA, arXiv 2022
> > >
> > > > Q2. Examples on "intentions part serve as the summary of interactions between ego agent and all other agents"
> > >
> > > We thank you for letting us further explain the building of the intention part. In our paper's Figure 1 (a), we have shown some examples, and we would like to show a complete example here. In one case, the interaction Q&As read:
> > >
> > > `[Q] What interactions are anticipated between the ego agent and surrounding agent #0..?`
> > >
> > > `[A] Surrounding agent #0 will yield to the ego agent because ..`
> > >
> > > `[Q] What interaction is likely to occur between the ego agent and surrounding agent #5?`
> > >
> > > `[A] Surrounding agent #5 will yield to the ego agent because ..`
> > >
> > > And, the intention Q&A summarize the interactions as followed:
> > >
> > > `[Q] What will the ego agent aim to do in the upcoming moments?`
> > >
> > > `[A] The ego agent intends to continue exiting the intersection. It will proceed as surrounding agents #0 and #5 are not moving and will yield to the ego agent. The other surrounding agents are also not moving and are not in the path of the ego agent, so no response is needed from the ego agent towards them..`
> > >
> > > This hierarchical structure is rooted in our prompts for building WOMD-Reasoning. In our paper's Table 8, the intention part of our dataset building prompt ask the GPT to: `think about following questions .. (1) What is the ego agent’s intention if no other agents exist? (2) .. what actions does the ego agent take to respond to traffic lights .. ? .. (3) In each interaction involving the ego agent, what actions do the ego and surrounding agent take to respond to each other? ..` Therefore the intention part would summarize the interactions and provide a comprehensive response.
> > >
> > > We hope this clarifies the structure of the intention part. For its impact, our goal for building WOMD-Reasoning is to provide an *auxiliary* driving model to offer explainability and introduce high-level knowledge. Our experiments in utilizing it for boosting trajectory predictions has proved the effectiveness in this sense. Also, we believe that this strategy is popularly adopted in LM-assisted driving models [2-4]. We will continue to update our dataset, including with human verifications, to make it even more reliable.
> > >
> > > Thanks again for helping make our paper better! We are happy to provide these additional details, and we kindly hope that our response solves your concerns!
> > >
> > > [2] Trajectory-LLM, ICLR 2025.
> > > [3] Large Language Models Powered Context-aware Motion Prediction in Autonomous Driving, IROS 2024.
> > > [4] Asynchronous Large Language Model Enhanced Planner for Autonomous Driving, ECCV 2024

---

### Official Review · Reviewer_LoCq · 2025-03-12

**Overall Recommendation:** 5

**Summary:**

The paper introduces WOMD-Reasoning, a large-scale, multi-modal dataset for reasoning about interactions in autonomous driving, focusing on traffic rule-induced and human intention-induced interactions—areas underrepresented in existing datasets. It also presents Motion-LLaVA, a multi-modal model fine-tuned on this dataset, demonstrating improved interaction prediction and traffic rule-compliant planning. Evaluations show strong performance gains over prior baselines.

**Claims And Evidence:**

The claims are well-supported.

**Essential References Not Discussed:**

The paper covers the most relevant work.

**Experimental Designs Or Analyses:**

The experimental design is sound. It would be improved by providing more details on the BEV input used for LLaVA and considering evaluations on real-world scenarios.

**Methods And Evaluation Criteria:**

The methods are appropriate:

1. Automated dataset generation with ChatGPT-4 and rule-based systems is clearly described.
2. Evaluations use standard metrics (BLEU, ROUGE, CIDEr, etc.) and human validation.

**Other Comments Or Suggestions:**

1. Some figures are cluttered, with overlapping numbers and diagrams that reduce readability. Simplifying the visuals would improve clarity and accessibility.
2. Consider running scaling studies to analyze how model performance evolves with different dataset sizes (e.g., 0.5M → 1M → 3M Q&A pairs). This would provide insights into the data efficiency of WOMD-Reasoning.
3. Include more comparisons with different backbone models (e.g., other LLMs or vision-language models) to better understand the advantages of Motion-LLaVA’s architecture and components.

**Other Strengths And Weaknesses:**

Strengths:

1. WOMD-Reasoning is the largest language-based dataset for autonomous driving interaction reasoning to date. It covers underexplored areas such as traffic rule-induced and human intention-induced interactions. Besides, despite being largely automatically generated, the dataset achieves around 90% accuracy in human evaluations, showing strong reliability for research purposes. The authors also provide the related code in the supplementary materials.
2. The provision of simulated BEV and ego-view videos adds versatility to the dataset, supporting vision-language research and training for autonomous systems.
3. The model fine-tuned on WOMD-Reasoning shows significant improvements in interaction prediction and traffic rule-compliant planning, key tasks for autonomous driving.
4. The application of CoT strategies reduces hallucination and improves reasoning quality, a practical enhancement for language-based models.


Weakness:
1. The paper focuses on WOMD data. It’s unclear how well Motion-LLaVA generalizes to other datasets or real-world cases without vectorized motion inputs.
2. The BEV inputs used in baseline comparisons with LLaVA are not well-detailed. Clarifying their preparation and limitations—especially in comparison to motion vectors—would be helpful.

**Questions For Authors:**

1. Clarify how agent IDs or numbers (e.g., surrounding agent #0, #1) are processed when fed into the network. Are they treated as categorical inputs, embedded, or pre-processed in another way?

**Relation To Broader Scientific Literature:**

The paper builds on prior datasets like BDD-X, DriveLM, and DRAMA, addressing their limitations in interaction reasoning. It advances the field by covering traffic rule-based and intention-driven interactions at scale.

**Theoretical Claims:**

No formal theoretical claims or proofs.

---

> ### Author Rebuttal · Authors · 2025-04-01
>
> We appreciate the reviewer's insightful feedback. Please see our response below.
>
> > Q1. Details on the BEV input used for LLaVA & Evaluations on real-world scenarios.
>
> We appreciate the reviewer's suggestions:
> 1) **For the BEV input**, it comes from plotting WOMD data. Road elements are plotted with line segments, while trajectories of agents are plotted with arrow-shaped boxes to indicate position and orientation. Trajectories are labeled with their Agent IDs. Additionally, an XY scale bar is added to help to interpret the position and the velocity.
> 2) **For real-world evaluations**, we first address that WOMD is a large real-world dataset. Therefore, testing our model on WOMD's cases can reflect the real world situations. To provide more evals on real-world tasks, we test introducing language outputs of Motion-LLaVA into trajectory prediction model Multipath++, seeing ~7% improvements on minFDE_6 and MR_6 in real-world WOMD cases. Details can be found in our answer to reviewer BaWk's Q2.
>
> > Q2. Generalization to other datasets or cases w/o motion inputs.
>
> We appreciate the question on generalizations.
> - **For other motion datasets**, our translation program convert motion data into raw language suitable for feeding LLMs to produce data in our dataset's format. With minor modifications in the translation program's IO part, it can be used for any motion dataset. Our translation program is included in the supplementary materials and will be open-sourced.
> - **For non-vectorized motion inputs**, we believe that the well-developed perception models can convert non-vectorized inputs into vectorized motion data. Furthermore, with our simulated visual modal, users can directly train VLMs with WOMD-Reasoning to take vision inputs.
>
> > Q3. BEV preparations and limitations
>
> We thank the reviewers for suggesting the comparison between BEV and motion vectors. The BEV preparations are detailed in Q1. BEV and motion vectors are equivalent representations of the same scenario. However, BEV images convey the coordinates and velocity info more implicitly, which poses challenges to extract these info. Therefore, our Motion-LLaVA choses motion data as inputs.
>
> > Q4. Simplifying the visuals.
>
> We appreciate the suggestions on simplifying the visuals to improve readability. We will fix this in the next version.
>
> > Q5. Data efficiency.
>
> We thank reviewer's suggestion on showing the data efficiency of WOMD-Reasoning by fine-tuning Motion-LLaVA with different data sizes. The results of the suggested experiments are listed below:
>
> For the **factual** questions:
> | Dataset Size | ROUGE (↑) | BLEU (↑) | METEOR (↑) | CIDEr (↑) | SPICE (↑) | GPT Score (↑) |
> |-|-|-|-|-|-|-|
> | 0.5M | 0.806 | 0.683 | 0.477 | 5.77 | 0.760 | 6.69 |
> | 1M | 0.818 | 0.701 | 0.491 | 5.90 | 0.773 | 6.74 |
> | 3M | 0.840 | 0.736 | 0.516  | 6.35 | 0.794 | 7.09 |
>
> For the **interaction reasoning** questions:
> | Dataset Size | ROUGE (↑) | BLEU (↑) | METEOR (↑) | CIDEr (↑) | SPICE (↑) | GPT Score (↑) |
> |-|-|-|-|-|-|-|
> | 0.5M | 0.562 | 0.410 | 0.339 | 1.94 | 0.513 | 5.67 |
> | 1M | 0.589 | 0.447 | 0.354 | 2.23 | 0.547 | 6.36 |
> | 3M | 0.614 | 0.474 | 0.366 | 2.52 | 0.571 | 6.76 |
>
> We find that with more WOMD-Reasoning data involved, the fine-tuned model first gains the ability to answer factual questions, which hits a good GPT score with only 0.5M data. As the dataset grows, the model gradually learns to answer interaction reasoning questions. These justify the size of WOMD-Reasoning, as well as its data efficiency in helping models to answer hierarchical driving-related questions.
>
> > Q6. Comparisons with different backbone models.
>
> We thank the reviewer for suggesting on using more LM baselines. We provide this comparison in our response to reviewer BaWk's Q1. Results show that w/o fine-tuning on WOMD-Reasoning, all baselines can hardly answer driving-related questions, supporting our dataset's motivation. We then fine-tune LLaVA and LLaMA-Adapter on WOMD-Reasoning, which both significantly benefit from the fine-tuning. Finally, we find that fine-tuned Motion-LLaVA works better than other fine-tuned models, supporting its superiority.
>
> > Q7. Processing of Agent IDs when fed into the network.
>
> We appreciate the question on agent IDs. These IDs are neither categorical inputs nor embedded. Instead, they are used as textual identifiers within the prompt, enabling the LLM to distinguish agents through natural language understanding. The prompt format is: `Ego agent: <motion>\nAgent #0: <motion>\n...\nNow, please answer: {Question}`, where  `<motion>` represents the encoded motion data from each agent's own viewpoint.
>
> These IDs only serve as range allocations, i.e. #0-#99 indicate vehicles, #100-#199 indicates bicycles and #200-#299 indicates pedestrians. We transform WOMD agent IDs to these local IDs to avoid downstream models overfitting to specific agent's behavior. Besides this, specific agent IDs do not carry any additional information.

---

### Official Review · Reviewer_TMNS · 2025-03-13

**Overall Recommendation:** 3

**Summary:**

The paper introduces WOMD-Reasoning, a large-scale dataset designed for interaction reasoning in autonomous driving, built upon WOMD. The dataset addresses a critical gap in understanding traffic rule-induced and human intention-induced interactions, which are often overlooked in existing driving datasets that primarily focus on proximity-based interactions. WOMD-Reasoning contains 3 million Q&A pairs spanning scene descriptions, motion predictions, and planning tasks for autonomous driving. To validate its effectiveness, the authors develop Motion-LLaVA, a motion-language model fine-tuned on WOMD-Reasoning, which demonstrates improved performance in interaction prediction and traffic rule-compliant planning.

**Claims And Evidence:**

The submission makes strong claims about WOMD-Reasoning as the largest multi-modal dataset for interaction reasoning in autonomous driving, with well-supported evidence from dataset comparisons, language model benchmarks, and fine-tuning evaluations of Motion-LLaVA.

However, some claims require stronger validation, particularly the high dataset accuracy. More details on evaluation criteria, distributional diversity, and human annotation reliability.

The claim that WOMD-Reasoning enables end-to-end traffic rule-compliant planning is not fully substantiated, as planning evaluations are mostly qualitative rather than tested in real-world driving or simulations. Additionally, while the dataset was automatically generated, its reliability and biases are not thoroughly analyzed.

To strengthen the paper, the authors should provide error analysis, real-world validation, and detailed evaluation metrics for reasoning quality and planning effectiveness.

**Essential References Not Discussed:**

There are some literatures that lack detailed  comparsion and discussion to outline the novelty:

[1] Sima, C., Renz, K., Chitta, K., Chen, L., Zhang, H., Xie, C., ... & Li, H. (2024, September). Drivelm: Driving with graph visual question answering. In European Conference on Computer Vision (pp. 256-274). Cham: Springer Nature Switzerland.

[2] Zhang, S., Huang, W., Gao, Z., Chen, H., & Lv, C. (2024). WiseAD: Knowledge Augmented End-to-End Autonomous Driving with Vision-Language Model. arXiv preprint arXiv:2412.09951.

**Experimental Designs Or Analyses:**

While the overall experimental design is appreciated, some issues remain.

There is a lack of systematic error analysis for the generated Q&As, particularly regarding potential (hallucinations, inconsistencies, and ambiguous cases) introduced by the automated pipeline. Without an in-depth breakdown of failure modes, it is unclear how reliable the dataset is for training and evaluating reasoning-based autonomous driving models. Motion-LLaVA’s failure cases are not analyzed, making it difficult to understand when and why the model makes incorrect predictions. A detailed error analysis of both the dataset and model outputs would provide stronger evidence of robustness and highlight areas for further improvement. Also, more transparent human evaluation is needed to ensure the reliability of the dataset and model outputs

**Methods And Evaluation Criteria:**

Proposed WOMD-reasoning generally make sense for the problem of interaction reasoning in autonomous driving, but there are some points could be improved:

1) Lack of real world/close loop evalutation: Using closed-loop evaluation, where the model's predictions influence simulated vehicle behavior, would better demonstrate its practical impact.

2) The claim that WOMD-Reasoning enhances traffic rule-compliant planning is mostly based on qualitative results. Metrics such as collision avoidance rate, rule violation rate, or trajectory efficiency in a planning framework could provide stronger evidence.

**Other Comments Or Suggestions:**

Refer to above.

**Other Strengths And Weaknesses:**

Refer to above.

**Questions For Authors:**

1. How do you ensure that the dataset accurately reflects real-world traffic interactions, especially for complex or rare cases?

2. What criteria were used to determine the correctness of Q&A pairs, and how was reasoning correctness evaluated?

3. How does Motion-LLaVA compare against non-language-based trajectory prediction or planning models in real-world performance?

**Relation To Broader Scientific Literature:**

The dataset and model enable better reasoning capabilities for language-driven autonomous driving systems, facilitating safer and more interpretable decision-making.

**Theoretical Claims:**

N/A.

---

> ### Author Rebuttal · Authors · 2025-04-01
>
> We thank the reviewer for the positive comments and insightful suggestions. Please see our responses below:
>
> > Q1. Dataset accuracy: Eval criteria, diversity and annotation reliability.
>
> We thank the reviewer for questions on validations, please see our responses below:
> 1) **Evaluation Criteria**: We perform 3 evals on WOMD-Reasoning and Motion-LLaVA fine-tuned on it:
>     - For human assessment of WOMD-Reasoning, the criteria is detailed in Q7.
>     - For evaluating the outputs of Motion-LLaVA, we use language metrics (ROUGE, BLEU, METEOR, CIDEr, and SPICE) along with a GPT Score which assesses semantic understanding beyond mere text similarity shown in Table 11.
>     - For evaluating WOMD-Reasoning in real-world applications, we add a trajectory prediction task, whose prediction metrics are minFDE_6 and MR_6.
> 2) **Distributional Diversity** WOMD-Reasoning's scenario diversity is backed by that of WOMD, a large-scale highly-interactive dataset. Also our paper's Figure 2 shows that among 63k scenarios covered in our dataset, there are ~120k `yields`, ~74k `lights`, ~74k `stops`, showing our dataset’s extensive coverage of traffic rule-induced interactions beyond just near-end events.
> 3) **Human Annotation Reliability** To justify the size of our human eval, we scale it up from ~1.6k to ~2.2k Q&As, and the correct rate remains stable (91.99% -> 91.80%), justifying the current eval scale.
> 4) **More Validations** We are tuning our data generation pipeline and incorporating WOMD's recent upgrades to offer new versions of the dataset with even better accuracy soon. We have also completed several human verification demos. We are proceeding to the human labeling.
>
> > Q2. Quantitative tests in real-world driving.
>
> We appreciate the questions on testing on the real world. First we address that testing Motion-LLaVA results on WOMD's val set can reflect real-world interactions, as WOMD is a real-world dataset. To provide more quantitative evals, we introduce language outputs of Motion-LLaVA into a trajectory prediction model Multipath++, seeing ~7% improvements on minFDE_6 and MR_6 in real-world WOMD cases. Details are included in our answer to reviewer BaWk's Q2.
>
> > Q3. Dataset's reliability and biases.
>
> We appreciate the question on reliability. We analyze the eval criteria, distribution diversity and human eval in Q1. Besides, in our response to reviewer BaWk's Q2, we show that WOMD-Reasoning helps a trajectory prediction model to enjoy ~7% improvements, which supports its overall reliability.
>
> > Q4. Error analysis.
>
> We appreciate the suggestions on error analysis. In our evaluations, we find that 1) Some scenarios are too complicated to describe in text. 2) Some errors still come from automatic translation program despite efforts. 3) We also see occasional error by the LM due to imprecise attention (e.g., confusing “following” when a car is behind but not in the same lane). Therefore, we are continuously updating and human-verifying the dataset to further improve its quality.
>
> > Q5. Evidence on traffic rule-compliance planning.
>
> We appreciate the suggestions. As the first step, we add a trajectory prediction task in our answer to reviewer BaWk's Q2, which proves overall effectiveness of our dataset on real-world tasks. We will follow the suggestions to quantify traffic rule-compliance metrics to give more specific evidence.
>
> > Q6. Literatures Comparison.
>
> We thank the reviewers for these important references. We discuss them below and will cite them in our next version.
> - **DriveLM**(ECCV 2024) We have cited and compared to it in Table 1 and in the introduction (line 28, right column). We will update its version in our next version.
> - **WiseAD**(ArXiv 2024) We see that WiseAD seems to be a driving model work rather than a new dataset. Their Tab. 1 lists the datasets they use for training. We hope our dataset will help encode traffic rule knowledge in models like WiseAD.
>
> > Q7. Criteria for evaluating Q&A pairs, and for reasoning.
>
> We appreciate questions on the eval criteria of Q&A pairs. Generally, we ask evaluators to judge whether the correct answer is included in the dataset's answers. Practically:
> For **facts**, the answer must be totally correct. (e.g. if number of lanes is asked, that number must be correct.)
> For **interaction reasoning**, the correct interaction or intention keyword must be included. (e.g., if human sees Agent #0 is required by rules to yield to Agent #1, then keywords `Agent #0 yields to Agent #1` must exist)
> These standards help us to maximize the eval accuracy while minimizing the human labor. We are performing human verifications to further boost data quality.
>
> > Q8. Motion-LLaVA v.s. non-language-based models.
>
> We thank the reviewer for the comparison request. However, instead of comparing, the outputs of Motion-LLaVA can be used as an addition to help real-world trajectory prediction models to perform better, like the ~7% improvement we observe (see our response to reviewer BaWk's Q2).

---

> > ### Comment · Reviewer_TMNS · 2025-04-04
> >
> > Thanks for the detailed response and additional exepriment against Multipath++. I will maintain my rating, with a positive inclination toward acceptance.

---

> > > ### Author Response · Authors · 2025-04-04
> > >
> > > We sincerely thank the reviewer for reading our response as well as acknowledging further evidence we provide in it. Also, we highly appreciate the reviewer's constructive suggestions which help us improve our work. Should you have any further questions, please let us know and we are more than happy to provide more!

---

### Official Review · Reviewer_BaWk · 2025-03-15

**Overall Recommendation:** 3

**Summary:**

This paper introduces the Waymo Open Motion Dataset-Reasoning (WOMD-Reasoning), a comprehensive question-and-answer dataset designed to articulate and assess the interactions prompted by traffic rules within driving scenarios. To demonstrate the utility of WOMD-Reasoning, the paper proposes Motion-LLaVA, a motion language model specifically fine-tuned using this dataset. The quantitative and qualitative evaluations confirms the dataset's quality and its usefulness in autonomous driving.

**Claims And Evidence:**

* This paper proposes a large-scale dataset with 3 million Q&A pairs centered on interaction reasoning in driving, which is valuable for the community if released.

**Essential References Not Discussed:**

* Table. 1 lists previous real-world language datasets for driving. However, some datasets like OmniDrive[4], NuInstruct[5], nuCaption[6], rank2tell[7], Tod3Cap[8] are not included.

[4] Wang, S., Yu, Z., Jiang, X., Lan, S., Shi, M., Chang, N., ... & Alvarez, J. M. (2024). Omnidrive: A holistic llm-agent framework for autonomous driving with 3d perception, reasoning and planning. arXiv preprint arXiv:2405.01533.

[5] Ding, X., Han, J., Xu, H., Liang, X., Zhang, W., & Li, X. (2024). Holistic autonomous driving understanding by bird's-eye-view injected multi-modal large models. In Proceedings of the IEEE/CVF Conference on Computer Vision and Pattern Recognition (pp. 13668-13677).

[6] Yang, S., Liu, J., Zhang, R., Pan, M., Guo, Z., Li, X., ... & Zhang, S. (2023). Lidar-llm: Exploring the potential of large language models for 3d lidar understanding. arXiv preprint arXiv:2312.14074.

[7] Sachdeva, E., Agarwal, N., Chundi, S., Roelofs, S., Li, J., Kochenderfer, M., ... & Dariush, B. (2024). Rank2tell: A multimodal driving dataset for joint importance ranking and reasoning. In Proceedings of the IEEE/CVF winter conference on applications of computer vision (pp. 7513-7522).

[8] Jin, B., Zheng, Y., Li, P., Li, W., Zheng, Y., Hu, S., ... & Zhao, H. (2024, September). Tod3cap: Towards 3d dense captioning in outdoor scenes. In European Conference on Computer Vision (pp. 367-384). Cham: Springer Nature Switzerland.

**Experimental Designs Or Analyses:**

* There are no comparison with baselines other than LLaVA in the manuscript. Some multi-modal VLMs could also achieve this, like Qwen[1], VITA[2], or llama-adapter[3]. As a dataset paper, the reviewer believes a through study of the baselines should be conducted.

[1] Bai, J., Bai, S., Chu, Y., Cui, Z., Dang, K., Deng, X., ... & Zhu, T. (2023). Qwen technical report. arXiv preprint arXiv:2309.16609.

[2] Fu, C., Lin, H., Wang, X., Zhang, Y. F., Shen, Y., Liu, X., ... & He, R. (2025). Vita-1.5: Towards gpt-4o level real-time vision and speech interaction. arXiv preprint arXiv:2501.01957.

[3] Gao, P., Han, J., Zhang, R., Lin, Z., Geng, S., Zhou, A., ... & Qiao, Y. (2023). Llama-adapter v2: Parameter-efficient visual instruction model. arXiv preprint arXiv:2304.15010.

**Methods And Evaluation Criteria:**

* This paper introduces Motion-LLaVA that provides interaction prediction for driving scenarios. The model utilizes a motion prediction model as a motion  encoder, followed by a LLaVA that generates language outputs.

**Other Comments Or Suggestions:**

None

**Other Strengths And Weaknesses:**

None

**Questions For Authors:**

* The method part primarily applies a motion encoder to replace the image encoder in LLaVA. Is there any adaptation specifically designed for the setting?

**Relation To Broader Scientific Literature:**

* As a language dataset designed for driving setting, it is essential to demonstrate its impact on downstream autonomous driving tasks like perception, motion prediction or trajectory prediction planning, and that's most of the existing research focusing on, like DriveLM[9] or Hint-ad[10]. Thus I recommend the authors conduct some experiments for the application of the dataset.

[9] Sima, C., Renz, K., Chitta, K., Chen, L., Zhang, H., Xie, C., ... & Li, H. (2024, September). Drivelm: Driving with graph visual question answering. In European Conference on Computer Vision (pp. 256-274). Cham: Springer Nature Switzerland.

[10] Ding, K., Chen, B., Su, Y., Gao, H. A., Jin, B., Sima, C., ... & Zhao, H. (2024). Hint-ad: Holistically aligned interpretability in end-to-end autonomous driving. arXiv preprint arXiv:2409.06702.

**Theoretical Claims:**

N/A

---

> ### Author Rebuttal · Authors · 2025-04-01
>
> We thank the reviewer for the positive comments and insightful suggestions, and we address your comments and concerns below.
>
> > Q1. More LM baselines.
>
> We are grateful for the suggestion. While our main goal for fine-tuning Motion-LLaVA is to show the effectiveness of WOMD-Reasoning, we agree that testing and fine-tuning more baselines would make our claims more reliable. Therefore, we test the quality of answers provided by a few vanilla and WOMD-Reasoning fine-tuned baselines, including LLaMA-Adapter, VITA and Qwen, all using 7B version. Results are shown below:
>
> | Model | Fine-tuned on WOMD-Reasoning | ROUGE (↑) | BLEU (↑) | METEOR (↑) | CIDEr (↑) | SPICE (↑) | GPT Score (↑) |
> |-|-|-|-|-|-|-|-|
> | LLaVA | ❌ | 0.512 | 0.211 | 0.275 | 1.36 | 0.455 | 2.31 |
> | LLaMA-Adapter v2.1 | ❌ | 0.413 | 0.174 | 0.235 | 0.91 | 0.372 | 1.62 |
> | VITA-1.5 | ❌ | 0.278 | 0.121 | 0.190 | 0.40 | 0.227 | 1.77 |
> | Qwen2.5-VL | ❌ | 0.384 | 0.156 | 0.247 | 0.62 | 0.379 | 2.36 |
> | LLaVA | ✅ | 0.779 | 0.581 | 0.439 | 5.51 | 0.735 | 6.88 |
> | LLaMA-Adapter v2.1 | ✅ | 0.722 | 0.470 | 0.375 | 4.72   | 0.691 | 5.14 |
> | Motion-LLaVA (Ours) | ✅ | **0.792** | **0.616** | **0.449** | **5.69** | **0.744** | **7.02** |
>
> We observe that w/o fine-tuning, all models can hardly answer driving-related questions, supporting the motivation of building WOMD-Reasoning. We then fine-tune LLaMA-Adapter on WOMD-Reasoning, which also significantly benefit from the fine-tuning. Due to limited resources, fine-tuning other baselines are still ongoing. Besides, we find that fine-tuned Motion-LLaVA works better than fine-tuned LLaVA or LLaMA-adapter, proving its well-designed structure in utilizing WOMD-Reasoning info.
>
> > Q2. Downstream Applications
>
> We thank the reviewer for this suggestion. To justify WOMD-Reasoning's ability in downstream tasks, we perform a trajectory prediction experiment to see the influence of using language outputs of Motion-LLaVA fine-tuned on WOMD-Reasoning. Multipath++ is used as the trajectory prediction baseline. As the interaction part is the most significant part in our dataset, we introduce the interaction part of language outputs into Multipath++, by cross-attending the T5 embedding of them with corresponding agents' features. Each experiment is run with 3 seeds for reliability. Averaged results are shown below,
>
> | Model | minFDE_6 (↓) | MR_6 (↓) |
> |-|-|-|
> | Multipath++ | 1.27 | 12.59% |
> | Multipath++ w/ Language | **1.18** | **11.69%** |
> | relative &Delta; | -7.35% | -7.10% |
>
> We observe a significant boost from language outputs, which strongly supports WOMD-Reasoning's ability to help downstream tasks like predictions. It can also help understanding the explainability of driving behaviors.
>
> > Q3. More comparisons.
>
> We thank the reviewers for these important papers to compare to! We will cite and compare to them, and we list a brief comparison here in our Table 1's form:
>
> | Dataset | Data Source | Total Scenes | Total Q&As | Interaction Q&As | Distance-induced | Traffic Rule-induced | Human Intention-induced | Scene Descriptions | Motion Prediction | Motion Planning |
> |-|-|-|-|-|-|-|-|-|-|-|
> | OmniDrive | nuScenes | <1k | N/A | N/A | ✅ | ✅ | ✅ | ✅ | ✅ | ✅ |
> | NuInstruct | nuScenes | 850   | 91k | <46k | ✅ | ❌ | ❌ | ✅ | ✅ | ✅ |
> | nuCaption | nuScenes | <1k | 420k | ~140k | ✅ | ❌ | ❌ | ✅ | ❌ | ✅ |
> | Rank2Tell | Rank2Tell | 116 | N/A | N/A | ✅ | ✅ | ✅ | ✅ | ✅ | ✅ |
> | Tod3Cap | nuScenes | 850 | ~2,300k   | 0 | ❌ | ❌ | ❌ | ✅ | ❌ | ❌ |
> | Ours | WOMD | **63k** | **2,940k** | **409k** | ✅ | ✅ | ✅ | ✅ | ✅ | ✅ |
>
> The stats further support that WOMD-Reasoning is the largest language Q&A dataset for driving by its release, with uniqueness in analyzing traffic rule-induced interactions.
>
> > Q4. The method part applies a motion encoder to replace the image encoder in LLaVA. Is there any adaptation?
>
> We appreciate the question on the design of Motion-LLaVA. Yes, we thoroughly adapt LLaVA to better encode motion info for answering driving-related questions, which are discussed in Appendix A.7. Besides introducing a motion encoder from Multipath++ to replace the vision encoder in LLaVA, we have also made following adaptations:
> 1) We use the encoder of a *pre-trained* Multipath++ as motion encoder. Motion prediction pre-training ensures the quality of the feature extracted from motion data, which is ablated in our Table 13.
> 2) We remove the projector alignment stage in LLaVA, and keep the motion encoder unfrozen in fine-tuning, to help the model better encode info for answering driving-related questions. The benefit of this is ablated in Table 14.
> 3) In Motion-LLaVA, we design a prompt for feeding motion embeddings from the Multipath++ encoder into LLM. The format is: `Ego agent: <motion>\nAgent #0: <motion>\n...\nNow, please answer: {Question}`. Here, `<motion>` represents the encoded motion data from each agent's ego-centric viewpoint. The Agent IDs are randomly assigned and serve only to differentiate agents in downstream Q&As.

---

> > ### Comment · Reviewer_BaWk · 2025-04-04
> >
> > The authors added comparisons with strong baselines (LLaMA-Adapter, Qwen, VITA), showing improvements after fine-tuning on WOMD-Reasoning. They also demonstrated the dataset’s utility in downstream trajectory prediction, and added comparisons to related datasets I previously mentioned. Clarifications on Motion-LLaVA’s design were also helpful. I remain WA.

---

> > > ### Author Response · Authors · 2025-04-04
> > >
> > > We sincerely thank the reviewer for reading our response as well as acknowledging further evidence we provide in it. Also, we highly appreciate the reviewer's constructive suggestions which help us improve our work. Should you have any further questions, please let us know and we are more than happy to provide more!

---

### Decision · Program_Chairs · 2025-05-01

**Decision:**

Accept (poster)

**Comment:**

This paper introduces the Waymo Open Motion Dataset (WOMD) of traffic rule-induced interactions when driving, with 3 million Q&A text annotations. The dataset is used to fine-tune a Motion-LLaVa model for predicting interactions in driving scenarios.

Reviewer BaWk, Qnzh , LoCq and TMNS praised the dataset contribution. Reviewer LoCq praised the addition of different view videos and of the fine-tuned WOMD-Reasoning model. Reviewer Qnzh praised the ablations.

Reviewers BaWk and TMNS suggested to include several references, baselines and downstream applications (which the authors did in the rebuttal). Reviewer TMNS also suggested more systematic error analysis. Reviewer LoCq requested some clarifications. Reviewer Qnzh had multiple questions on which they engaged with the authors.

Given the the average score of 3.5 I recommend acceptance.